# Designed active-site library reveals thousands of functional GFP variants

Jonathan Yaacov Weinstein [1], Carlos Martí-Gómez[2], Rosalie Lipsh-Sokolik [1], Shlomo Yakir Hoch [1], Demian Liebermann[3], Reinat Nevo[1], Haim Weissman[4], Ekaterina Petrovich-Kopitman[5], David Margulies [6], Dmitry Ivankov [7], David M. McCandlish[2] & Sarel J. Fleishman [1] ✉

Mutations in a protein active site can lead to dramatic and useful changes in protein activity. The active site, however, is sensitive to mutations due to a high density of molecular interactions, substantially reducing the likelihood of obtaining functional multipoint mutants. We introduce an atomistic and machine-learning-based approach, called high-throughput Functional Libraries (htFuncLib), that designs a sequence space in which mutations form low-energy combinations that mitigate the risk of incompatible interactions. We apply htFuncLib to the GFP chromophore-binding pocket, and, using fluorescence readout, recover >16,000 unique designs encoding as many as eight active-site mutations. Many designs exhibit substantial and useful diversity in functional thermostability (up to 96 °C), fluorescence lifetime, and quantum yield. By eliminating incompatible active-site mutations, htFuncLib generates a large diversity of functional sequences. We envision that htFuncLib will be used in one-shot optimization of activity in enzymes, binders, and other proteins.

Protein active sites comprise molecular-interaction networks that are critical to function. Due to the molecular density of the active site, however, the majority of mutations destabilize the protein[1] or lead to dysfunction[2], and functional multipoint mutants are exceptionally rare[3,4]. Thus, active sites are among the most evolutionarily conserved protein sites[5]. Furthermore, experimental lab-evolution studies that aim to modify protein activity typically discover many more mutations outside the active site than within it[6]; yet, understanding whether and how remote mutations change activity is often elusive[7,8]. Although active-site mutations have the greatest potential to alter function, in practice, sensitivity to mutation has severely limited access to active-site functional variants in natural and lab evolution, deep mutational scanning[9,10], and computational protein design[11]. Therefore, as a rule, lab-evolution studies comprise multiple cycles of mutagenesis and

selection that are customized specifically for each desired functional trait[12–14]. Such iterative processes are time consuming and likely to severely undersample the space of functional sequences.

Furthermore, epistatic interactions between mutations can severely restrict the chances of finding functional multipoint mutants[15]; that is, a mutation may be tolerated only in combination with one or more additional mutations[16–18], drastically reducing the chances for the emergence of beneficial multipoint mutants[15,19]. This dependence also severely limits our ability to predict the functional impact of multipoint mutations even when the effects of single-point mutations are known[20,21], for instance, based on deep mutational scanning[3,4]. Epistasis has critical implications for our understanding of molecular evolution, including the emergence of viral and microbial resistance mutations[22] and the evolution of new enzymatic and binding

[1]Department of Biomolecular Sciences, Weizmann Institute of Science, Rehovot 7610001, Israel. [2]Simons Center for Quantitative Biology, Cold Spring Harbor Laboratory, Cold Spring Harbor, NY 11724, USA. [3]Department of Chemical and Biological Physics, Weizmann Institute of Science, Rehovot 7610001, Israel. [4]Department of Molecular Chemistry and Materials Science, Weizmann Institute of Science, Rehovot 7610001, Israel. [5]Life science Core facilities, Weizmann Institute of Science, Rehovot 7610001, Israel. [6]Department of Chemical and Structural Biology, Weizmann Institute of Science, Rehovot 7610001, Israel. [7]Center of Life Sciences, Skolkovo Institute of Science and Technology, Moscow, Russia. ✉e-mail: sarel@weizmann.ac.il

specificities[23]. It also presents one of the primary obstacles to our ability to design protein activities in basic and applied research[1,24].

Here, we introduce a computational method called high-throughput Functional Libraries (htFuncLib) to design large libraries of active-site mutants that can be applied, in principle, to any protein. Most current atomistic design methods, including our previously described FuncLib method[24], select designs that optimize desired energy or structure criteria[25,26]. By contrast, htFuncLib searches for a set of active-site point mutations that, when freely combined, yield low-energy multipoint mutants. Our approach can be applied to an arbitrarily large set of positions to generate diverse and complex libraries that encode millions of designs. htFuncLib thus accesses sequence spaces that have so far been interrogated through random or semi-random mutagenesis and selection methods. Yet, unlike such methods, htFuncLib generates libraries that are preselected computationally to enrich for stable, folded, and potentially active designs.

## Results

### Principles for designing combinatorial active-site diversity

We applied htFuncLib to Green Fluorescent Protein (GFP). GFP and other fluorescent proteins have attracted intense interest in evolution studies due to their ubiquitous uses in molecular and cellular biology[27–29] and their straightforward optical readout[30]. GFP fluorescence depends on the chemical environment of the chromophore, including electrostatics and torsional freedom about the bond that links its aromatic rings[31] and is therefore sensitive to mutations in the chromophore-binding pocket. Most previous large-scale screens targeted the entire protein or consecutive segments of it for mutation[3,4,30,32]. GFP is a β-barrel, however, and the chromophore is buried within the protein core. Therefore, most mutations targeted solvent-exposed regions that are unlikely to impact spectral properties. Unlike these previous studies, we apply htFuncLib solely to positions that line the chromophore-binding pocket. Because active-site mutations may reduce protein stability, we chose as a starting point a previously designed version of enhanced GFP, PROSS-eGFP, that exhibited elevated resistance to thermal denaturation[33]. In this previous design, positions in the chromophore-binding pocket, except Tyr145Phe and Thr167Ile, were immutable. In applying htFuncLib, we also allowed design in these two positions.

Our working hypothesis is that epistatic interactions most frequently arise from three molecular sources (Supplementary Fig. 1): (1) direct molecular interactions between proximal mutated amino acids; (2) indirect interactions between amino acid positions due to backbone conformational changes; and (3) stability-mediated interactions in which destabilizing mutations do not exhibit phenotypic differences when introduced singly but reduce stability or expression levels when combined[1,7]. htFuncLib addresses these sources of uncertainty in designing multipoint combinatorial mutants, as described below.

The htFuncLib approach combines phylogenetic analysis, Rosetta atomistic design calculations[26,34], and a machine-learning analysis to nominate mutations that are mutually compatible when combined freely with one another (see "Methods" for details). Using Fig. 1 as a visual guide for applying htFuncLib to GFP, we started by manually selecting 27 active-site positions likely to impact functional properties based on previous studies of GFP or proximity of these positions to the chromophore (Fig. 1A). htFuncLib then computed all single-point mutations and selected the ones likely to be tolerated against the background of the original amino acids in all other positions[34]. In this

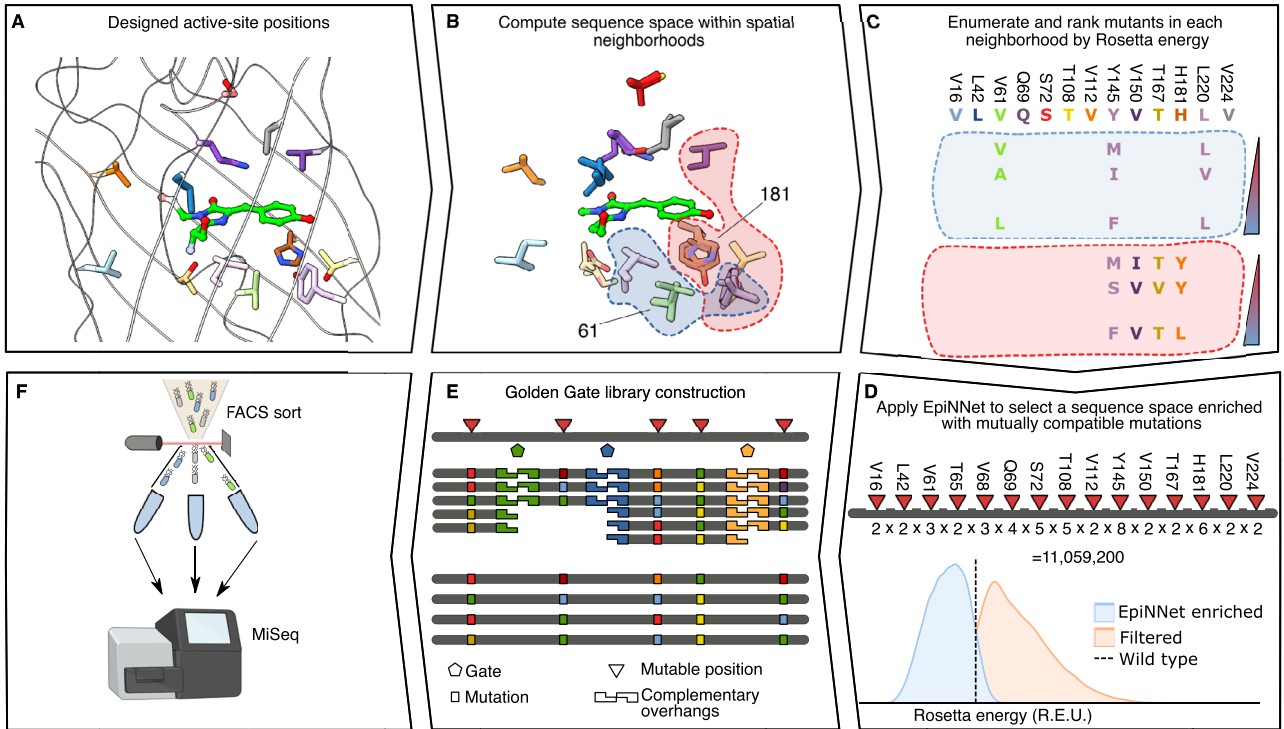

**Fig. 1 | Steps in applying htFuncLib to GFP. A** Fourteen positions designed by htFuncLib are shown (PDB entry: 2WUR). **B** Red and blue backgrounds indicate representative neighborhoods centered around GFP amino acid positions 181 and 61, respectively. **C** The sequence space of each neighborhood is partially enumerated. Sequence representation of the two neighborhoods shown in (**B**). Only variable positions are shown for clarity. Color bars represent Rosetta energies. **D** EpiNNet top-ranked mutations are selected as the enriched sequence space. An atomistic verification step scores thousands of random combinations from the EpiNNet-enriched and the filtered sequence spaces. Nearly all designs in the EpiNNet sequence space are predicted to be more stable than PROSS-eGFP, compared to almost none in the filtered sequence space. Red triangles mark mutable positions, and the number of mutations in each position is marked under the bar. **E** The designed library is cloned using Golden-Gate assembly[38] of oligos that contain the desired mutations, expressed in *E. coli* cells, and **F** sorted by FACS. Icon created using BioRender.

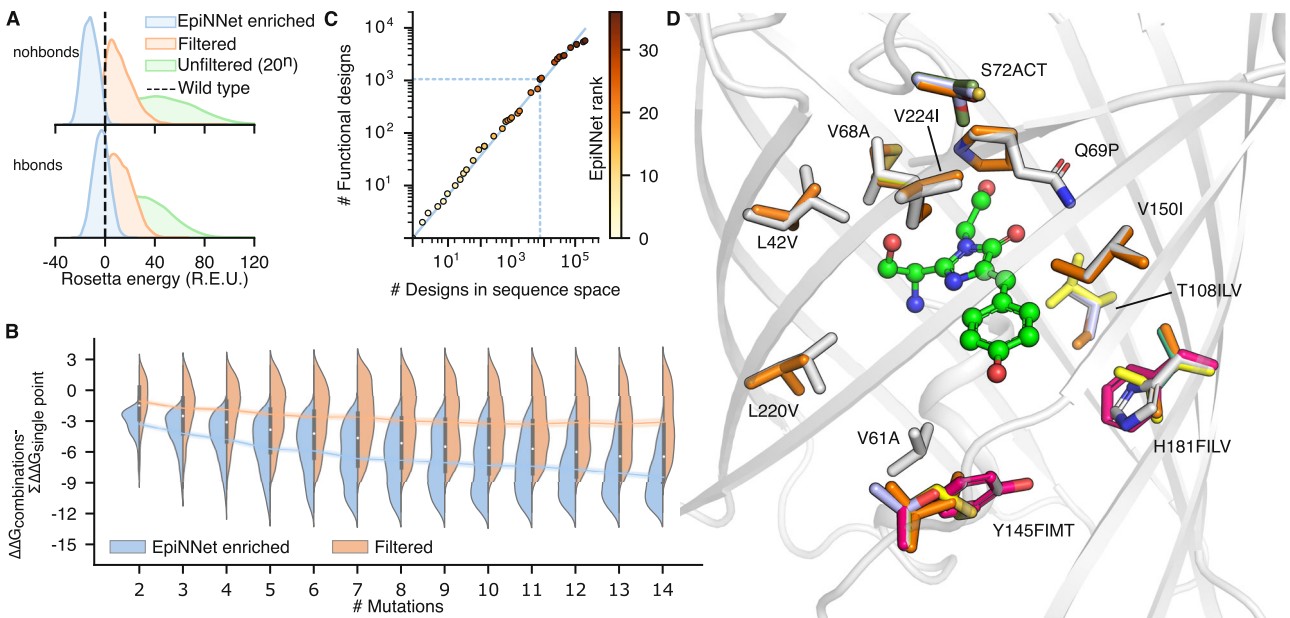

**Fig. 2 | htFuncLib selects mutations that combine to form low-energy designs.**
**A** Energy distributions of the EpiNNet-enriched sequence space, the sequence space filtered by energy and phylogenetic criteria (Filtered), and unfiltered (all 20 amino acids at each position). >95% of mutants in the EpiNNet-enriched combinatorial sequence space exhibit higher stability than PROSS-eGFP, compared to <0.6% for the other spaces. 12,000 randomly selected sequences were modeled to generate each distribution. The dashed line signifies PROSS-eGFP energy.
**B** Distributions of the energy difference between multipoint mutants and the sum of their constituent point mutations. Up to 1000 random combinations of mutations were modeled and scored for each distribution. Box bounds and the white circle signify the first, second and third quartiles. Whiskers represent 1.5 times the inter-quartile range. **C** The number of functional designs according to FACS screening of libraries comprising an increasing number of top-ranked EpiNNet mutations is plotted as a function of the total number of designs detected in the deep-sequencing data. Points are color-coded according to the number of mutations that constitute the library. For example, a library of 25 top-ranked EpiNNet mutations that comprise ~$10^4$ designs would yield approximately $10^3$ functional ones (dashed blue lines). The diagonal is the best fit to the data points. **D** Overlay of all mutations of the 25 top-ranked EpiNNet mutations from (**C**). Despite the relatively small size of this library, it contains radical mutations, including Tyr145Met and Gln69Pro. Data are provided as a Source Data file.

selection step, we retain mutations that are likely to be present in the diversity of sequence homologs and that are moreover predicted not to destabilize the native state according to atomistic design calculations[35]. The atomistic calculations contain the chromophore to ensure that the mutations do not abrogate contacts that may be critical to fluorescence. In addition, these calculations apply harmonic coordinate constraints to backbone atoms during whole-structure minimization, thereby penalizing backbone deformations that may lead to indirect epistatic interactions (Supplementary Fig. 1B).

After filtering, htFuncLib applies atomistic modeling to evaluate the energy of combinations of tolerated point mutations. Because complete enumeration of the space of potential multipoint mutations in a large active site is computationally intractable, we focus calculations on combinations of mutations within neighborhoods of proximal positions (Fig. 1B, C, Supplementary Datasets 1 and 2) which are the most likely to give rise to direct epistatic interactions (Supplementary Fig. 1A). In a companion paper, we show how to select combinations of enzyme backbone fragments that form low-energy combinations when freely combined using a machine-learning-based approach called EpiNNet[36]. Here, we extend EpiNNet to select low-energy combinations of mutations across all spatial neighborhoods within the chromophore-binding pocket. The multipoint mutants within each neighborhood are classified according to their energies into favorable (Rosetta energies lower than PROSS-eGFP) and unfavorable (highest-energy 50%, Fig. 1D). We then train the neural network to predict the energy-based classification of favorable and unfavorable designs. Finally, the trained network ranks the single-point mutations according to their likelihood to be found in low-energy multipoint mutants, and the top-ranked mutations are selected for library construction. EpiNNet comprises a single fully connected hidden layer. Therefore,

unlike some current linear-regression based techniques for predicting favorable multipoint mutants[37], EpiNNet can model some of the nonlinear relationships between mutations that determine the energy outcome. Such nonlinear interactions may be dominant in a highly epistatic region such as an active site. The resulting library is enriched in mutually compatible (low-energy) mutations, such that both direct and stability-mediated epistasis (Supplementary Fig. 1A, C) are addressed. Following design, we clone the library using Golden-Gate assembly[38] (Fig. 1E) and apply FACS sorting and deep sequencing to identify active designs (Fig. 1F).

Multipoint mutants from the EpiNNet-enriched sequence space exhibit, on average, dramatically lower computed energies than those in the original filtered sequence space (Fig. 2A), suggesting that EpiNNet increases the fraction of folded and stable designs. Furthermore, in a representative case, following the selection steps, only 14 positions (out of the 27 we selected initially) were selected for design with a sequence space of $10^7$, compared to experimentally intractable $10^{35}$ sequences for the space encompassing every mutation at 27 positions and $10^{19}$ following the phylogenetic and single-mutation energy filters (Supplementary Datasets 3 and 4).

Thus, unlike conventional protein design methods[25,26], htFuncLib does not search for the most optimal mutants according to energy or structural criteria. Instead, the astronomically large space of combinatorial mutations in an active site is reduced to a tractable size through phylogenetic, structural, and energy-based analysis. Then, mutations that may destabilize the protein in combination with others are removed by analyzing the energies of combinatorial mutations. htFuncLib assumes that active-site stability is a primary constraint for discovering functional multipoint mutants[1,39,40]. Additional functional constraints are encoded by verifying that the mutants form favorable interactions with the chromophore.

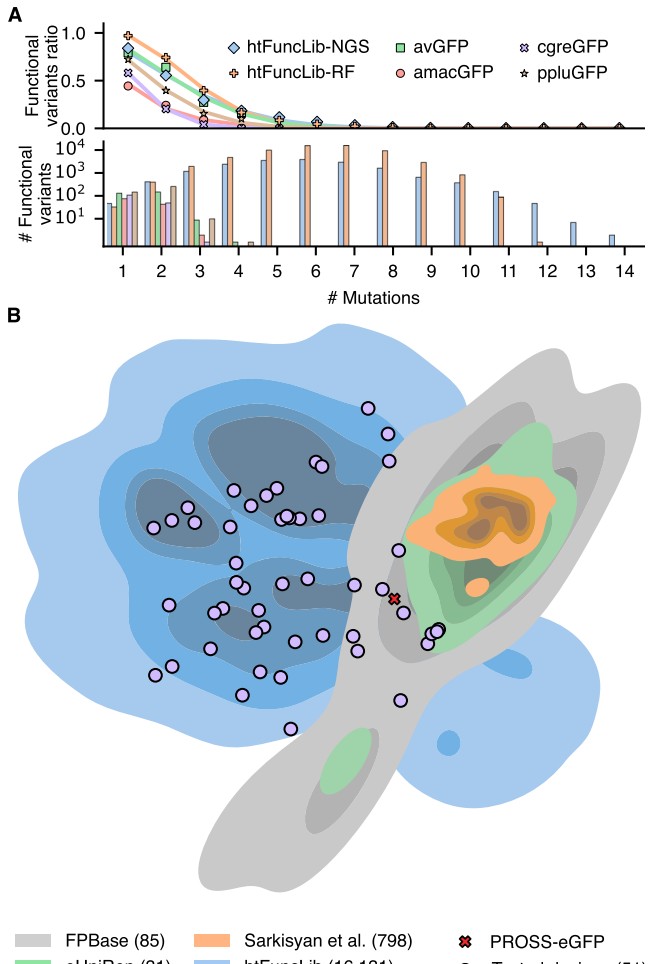

**Fig. 3 | htFuncLib exposes a large space of functional multipoint active-site GFP variants.** Deep sequencing of htFuncLib libraries sorted by fluorescence revealed over 16,000 potentially active designs. **A** Frequency and number of functional variants with a given number of mutations (top and bottom, respectively). htFuncLib-NGS - all sequences obtained from deep sequencing of the sorted designs; htFuncLib-RF - the entire sequence space labeled by the random forest. The avGFP dataset was derived from Sarkisyan et al.[4]. The amacGFP, cgreGFP, and ppluGFP datasets were derived from Somermeyer et al.[3]. Lines represent fits to the data (points) according to Eq. 2 (see "Methods" and Supplementary Table 2). Data excluded sequences with mutations outside of the chromophore pocket. **B** Distance-preserving dimensionality reduction analysis shows the relationships between GFP variants in FPBase[35], Sarkisyan et al.[4], eUniRep[41], and htFuncLib. The plot approximates the number of mutations between any pair of mutants[41,75]. PROSS-eGFP (and eGFP, which are nearly identical in the designed positions, Supplementary Dataset 7) are marked by a cross for reference. Individually characterized htFuncLib designs are marked by purple circles. The number of sequences represented for each category is marked in parentheses. Variants with mutations outside the chromophore pocket were included, but these mutations were ignored when calculating distances. Data are provided as a Source Data file.

## Design of a multiplexed GFP active-site library

The spectral properties of GFP depend on chromophore packing, electrostatics, and hydrogen-bond networks around the chromophore[27]. Since hydrogen-bond networks are extremely sensitive to structural perturbations, we designed two libraries: nohbonds, which excluded positions that directly hydrogen bond to the chromophore, and hbonds, which included such positions. We manually selected 27 and 24 positions for design in each library, respectively, applied htFuncLib to these positions, and generated 11 million and 930,000 designs for each library, respectively. Both libraries are complex: some positions allow only subtle mutations, and others,

including e.g., Gln69 and Tyr145, exhibit high diversity and radical mutations (Supplementary Figs. 2 and 3, Supplementary Dataset 5). According to Rosetta atomistic modeling, both libraries are highly enriched for low-energy mutants compared to the GFP starting point. For instance, nearly 99% and more than 67% of the nohbonds and hbonds designs, respectively, exhibit lower Rosetta energies than the progenitor PROSS-eGFP (Fig. 2A). By contrast, the energies of multi-point mutants from the sequence space prior to EpiNNet enrichment are significantly worse than PROSS-eGFP, with >99% and >96% exhibiting higher energies for nohbonds and hbonds, respectively (Fig. 2A). The unfavorable energies of combinatorial mutants in the sequence space before EpiNNet selection reflect the high epistasis in the active site. By contrast, the EpiNNet-enriched sequence space significantly improves the fraction of low-energy and, thus, potentially stable and foldable active-site designs. Additionally, combinations of EpiNNet-selected mutations exhibit lower energies than expected from an additive contribution of the constituting point mutations (Fig. 2B).

The two libraries were cloned using Golden-Gate assembly into *Escherichia coli* cells, with transformation efficiency greater than $5 \times 10^7$. Deep-sequencing analysis of the unsorted libraries shows high uniformity in the distribution of multipoint mutations, verifying that the assembly process exhibits low bias (Supplementary Fig. 4). The cells were FACS-sorted using two selection gates: (405 nm excitation, 525 nm emission; referred to as AmCyan[405/525]) and (488, 530 nm; referred to as GFP[488/530]; Supplementary Fig. 5). Following selection, plasmids were purified and cloned into fresh cells and resorted using the same gating strategy to reduce sort errors. Following each sort, we collected several individual clones for sequencing and functional measurements, obtaining 62 unique designs, 50 of which were functional. Furthermore, the presorted library and the output from the second sort were subjected to deep-sequencing analysis. To determine thresholds for selecting positive hits from the deep-sequencing data, we analyzed the enrichment values of the 62 designs we collected during sorts. Relatively loose criteria (enrichment in the selected population relative to the presorted population >1) captured 45 functional designs with only a single false positive (Supplementary Table 1). Applying these thresholds, we identified 14,242 and 1926 unique designs in the sorted nohbonds and hbonds libraries, respectively (0.13% and 0.21%, respectively; see Supplementary Fig. 6 for distribution of read counts in the selected libraries). We also retrospectively evaluated the fraction of functional GFP variants in libraries that were constructed from top-ranked EpiNNet-selected mutations. We found that up to library sizes of $10^4$–$10^5$, approximately 10% of the multipoint mutants were functional, and only above a library size of $10^5$ did the fraction of functional variants decay substantially (Fig. 2C, D). These results are encouraging as they suggest that focusing htFuncLib on top-ranked mutations may yield highly functional libraries in experimental systems that are not amenable to high-throughput screening.

Combining the positive hits from both libraries yields 16,155 unique, putatively active GFP designs. These include 1167 designs that exhibit ≥8 mutations relative to GFP (Fig. 3A). Many of the active designs exhibit radical mutations, including Thr203His (13%), Gln69Met (9%), Ser205Asp (9%), Gln94Leu (8%), and Tyr145Met (8%) (Supplementary Dataset 6). The large number of functional active-site multipoint mutants is striking compared to previous engineering and design strategies applied to eGFP, which showed a steep decline in active mutants with the number of mutations and no active mutants with ≥5 mutations in the chromophore-binding pocket[4,41] (albeit, these studies did not focus diversity on the active site). The vast majority of the mutations observed in those studies were in the more tolerant solvent-exposed surfaces. By contrast, the current designs are entirely within the chromophore-binding pocket where they are more likely to affect functional properties (Fig. 3A). The large number of active high-

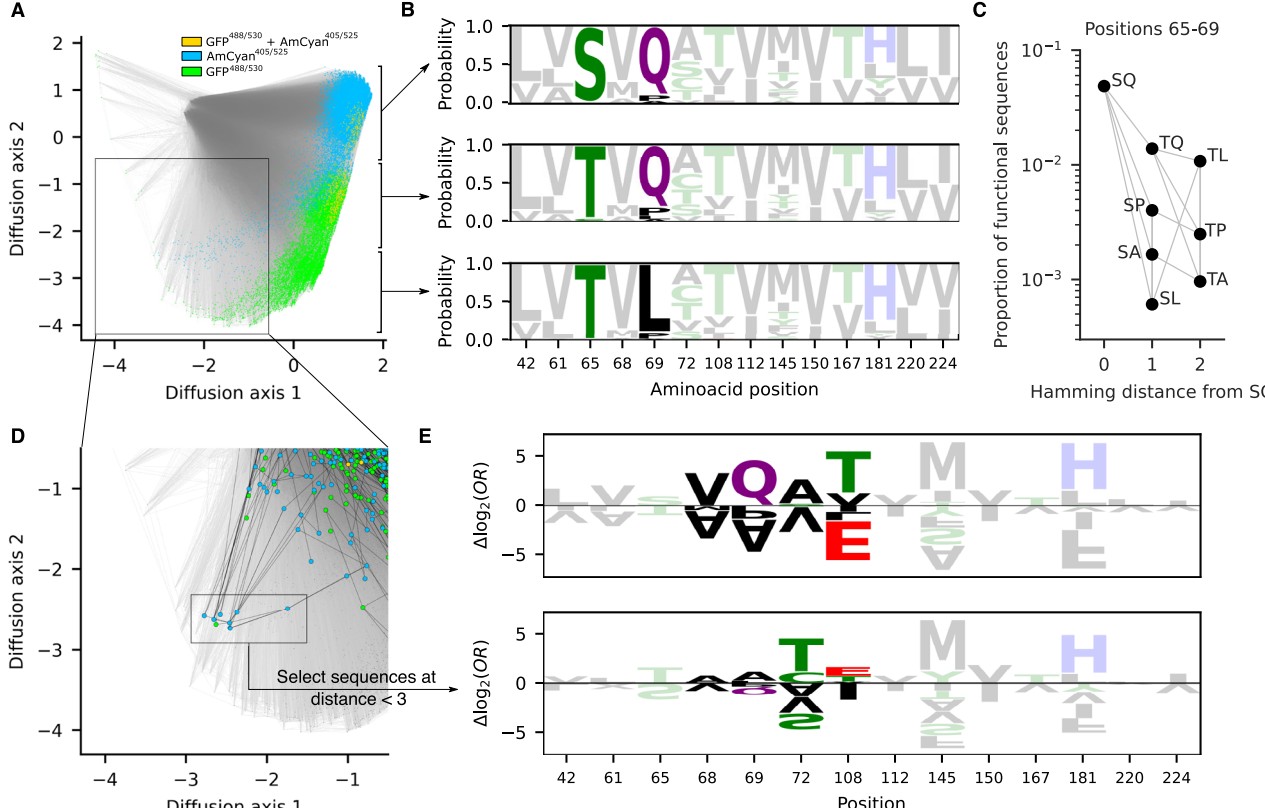

**Fig. 4 | Global analysis of the GFP genotype-phenotype map shows high mutational contiguity among functional sequences. A** Low-dimensional visualization of the sequence-function relationship predicted by the random forest model (see "Methods"). Functional sequences are highlighted in different colors according to whether they are predicted to fluoresce in the GFP488/530 channel (green), AmCyan405/525 channel (blue), or both (gold). Lines join genotypes that are separated by a single amino acid substitution. **B** Site-frequency logos of functional sequences based on position along diffusion axis 2 (the three logos correspond to diffusion axis 2 coordinates greater than −0.5, between −0.5 and −2.25, and less than −2.25). **C** The proportion of functional sequences changes depending on the amino acids at positions 65 and 69. Gray lines indicate single amino acid substitutions. **D** Close-up of the region containing a cluster of observed sequences with unusual sequence properties. Highlighted dots indicate sequences that were directly characterized as functional in the high-throughput experiments, and black lines indicate single amino acid substitutions between these experimentally characterized sequences (see Supplementary Fig. 9 for a visualization of all sequences enriched in the high-throughput experiment). **E** Sequence logo representing the coefficients of the logistic regression models trained on random forest predictions to identify changes in allelic preferences when using all sequences for training (top) or only sequences within two mutations of the genotypes highlighted in (**D**) (bottom). Coefficients are expressed as additive allelic contributions (i.e., Δlog₂ odds ratios) that have been mean-centered by site. Data are provided as a Source Data file.

order multipoint designs in our dataset confirms our working hypothesis that a stable starting scaffold (eGFP-PROSS) and the htFuncLib enrichment of mutually compatible mutations dramatically increase the yield of functional active-site multipoint mutations. Furthermore, htFuncLib generates many more functional multipoint active-site designs relative to random mutagenesis (Fig. 3A). Finally, compared to all known descendants of *Aequorea victoria* GFP (avGFP) in the fluorescent protein database (FPBase)[35] and variants characterized in focused and high-throughput studies, htFuncLib explored different regions of the sequence space (Fig. 3B).

**Random forest modeling of GFP genotype-phenotype map**
To gain insight into what determines the functional outcome of multipoint mutants in the htFuncLib designs, we trained a random forest model using the functional annotations derived from the deep-sequencing data for the nohobnds library. We chose this type of analysis because it is easily interpretable, less prone to overfitting than other approaches, and well suited for mixed categorical and numerical data. As features for training, we used the mutation identities, geometric and physicochemical properties, and conservation scores. The best-performing model exhibits 84% accuracy in predicting functional versus non-functional designs in a balanced test (Supplementary Fig. 7,

Supplementary Table 3). The most important single feature for predicting functionality is the mean conservation score, calculated as the sum of differences in the conservation scores between PROSS-eGFP and mutated identities (ΔPSSM, Supplementary Fig. 8). In fact, this single parameter exhibits an area under the ROC curve of 87%, compared to 93% for the random forest. This result provides a compelling verification for the approach of combining sequence conservation with atomistic protein design which underlies htFuncLib and other successful protein design methods developed in recent years[26].

To further understand the qualitative features of the sequence-function relationship learned by the random forest, we used a technique for visualizing complex fitness landscapes[42]. In this technique, the distance between sequences reflects the time it would take for a population to evolve from one sequence to another under selection to maintain a fluorescent phenotype as predicted by the random forest model (see "Methods"). We found that the main structure of the landscape could be represented by a two-dimensional visualization, where each axis captures a different qualitative feature of the GFP genotype-phenotype map (Fig. 4A). The first axis (diffusion axis 1) mainly distinguishes functional from non-functional sequences (79% of sequences with diffusion axis 1 values greater than 1 were functional, while only 0.01% were functional if they had diffusion axis 1 values less

than −1), capturing the fact that the functional sequences are highly connected with each other and localized in sequence space rather than consisting of isolated fitness peaks separated by valleys. The contiguity between functional sequences suggests that the htFuncLib selection of mutations that increase stability may generate a highly evolvable library in which active variants are connected via mutational trajectories that maintain function[39,40]. Additionally, the second axis (diffusion axis 2) then largely separates functional AmCyan[405/525] sequences from functional GFP[488/530] sequences.

Figure 4B provides more detail on the interpretation of diffusion axis 2 by showing site-frequency logos for three different regions of the fitness landscape. These frequency logos indicate that the main set of functional sequences is largely separated into three groups: one group with Thr65Ser and Gln69 consisting of AmCyan[405/525] designs; one group with Thr65 and Gln69 consisting of designs that fluoresce a mixture of AmCyan[405/525], GFP[488/530], or both; and one group with Thr65 and Gln69Leu that consists of GFP[488/530] designs. All three groups are strongly supported by many different sequences directly assessed in the sorting experiment (Supplementary Fig. 9). Strikingly, Thr65Ser and Gln69Leu are highly incompatible: sequences that contain both these mutations have a much lower chance of being functional (Fig. 4C). As a consequence, evolutionary trajectories from Ser65/Gln69 to Thr65/Leu69 that maintain functionality would tend to pass through a Thr65/Gln69 intermediate.

In addition to these three main groups, which comprise the large majority of the functional designs, the random forest analysis predicts two long parallel tails of functional sequences spreading along diffusion axis 1 and sweeping up along diffusion axis 2. The AmCyan[405/525] tail is well-supported by the experimental data and is not an artifactual prediction of our model, as we observe a cluster of highly mutationally connected designs that were also among the most strongly enriched in AmCyan[405/525] sorted cells (Fig. 4D, Supplementary Fig. 9, Supplementary Table 4). Moreover, in this cluster, all sequences contain an unusual and rarely functional pairing of alleles Thr65/Gln69Ala (Fig. 4C), and all except one contain Thr108Glu, which is also unusual among other functional sequences (Fig. 4B). To investigate what distinguishes these designs from the other fluorescent proteins in the library, we fit an additive logistic regression model to the random forest output using only sequences up to two mutations away from the cluster highlighted in Fig. 4D (see "Methods"). We then compared the estimated mutational effects on the probability of activity to those obtained by fitting the same logistic regression model to the full genotypic space (Fig. 4E). Although there are some commonalities in the inferred mutational effects (e.g., Tyr145Met, which is the strongest single-site predictor of functionality based on the random forest, greatly increases the probability that a sequence is fluorescent under both logistic regression models), positions 68, 69, 72 and 108 show marked differences in amino acid preferences. For example, Thr72Ala increases the odds ratio for functionality by approximately fourfold in the general model but reduces the odds ratio by 13-fold in this alternative context. These results suggest that variants within this cluster also differ in their functional constraints as compared to the majority of fluorescent designs, although more detailed experiments would be required to validate this qualitatively different solution to GFP fluorescence.

## Designs exhibit large and useful functional diversity

The above results, based on flow cytometry, identify designs that maintain fluorescence, but they do not provide information on other changes in functional properties, including finer-scale changes in excitation and emission spectra. To examine these aspects of functional diversity, we expressed, purified, and characterized a total of 88 unique designs, exhibiting at least two mutations from PROSS-eGFP and typically at least two mutations from one another, and three controls (eGFP, PROSS-eGFP, and superfolder GFP (sfGFP);

Supplementary Dataset 7). Twenty-four designs are cluster representatives of the hits observed in the deep-sequencing data, 17 of which (71%) were active. We also selected three designs with mutations rarely found in the sorted populations, Glu222Gln/Leu and Leu44Met, one of which was active. As an especially stringent test, we selected six designs with the maximal number of mutations (12–14), but none of these was functional. Furthermore, we selected 19 designs that were predicted to be functional by the random forest analysis but were not observed among the positive hits in the deep-sequencing analysis. Surprisingly, 15 (79%) were active, confirming that a random forest analysis based on deep-sequencing data of htFuncLib designs can be used to recover false negatives—active designs that were missed by the experimental workflow. Additionally, we isolated designs from FACS sorts that were gated for higher brightness or spectral shifts (Supplementary Table 5) by applying sorting gates that combine two channels (Supplementary Fig. 10). We also verified that 19 designs could be transferred to the superfolder GFP (sfGFP) background[43] to demonstrate that the designs are compatible with a different chassis (Supplementary Dataset 7).

Although we did not explicitly guide the design process to improve any functional property (except native-state stability (Figs. 1D and 2A)), we hypothesized that the large diversity in active-site sequences would lead to observable functional differences. We first analyzed GFP functional thermostability or the temperature at which its fluorescence deteriorates to 50% of the maximal value, a critical property for high-temperature or long-term experiments and "real-world" applications[44,45]. Functional thermostability is remarkably variable among the designs, 46–96 °C, compared to 84 °C for eGFP (Fig. 5A and Supplementary Fig. 11). We noticed that the PROSS-eGFP parental design is less stable than eGFP when functional thermostability is measured (Fig. 5A) rather than thermal denaturation as in the PROSS-eGFP design study[33]. Apparently, the PROSS-eGFP design is more resistant to heat denaturation, but its fluorescence is more sensitive to heat than eGFP. Quantum yield, which measures the efficiency of emitting light absorbed by the chromophore, was also variable, ranging between 0.16–0.82, compared to 0.55 for eGFP (Fig. 5A and Supplementary Fig. 12). Surprisingly, across all the designs we tested, functional thermostability and quantum yield were correlated (Pearson's $r = 0.53$, Supplementary Fig. 13). This correlation may stem from the fact that both chromophore brightness and resistance to unfolding increase with core packing density[31,46]. To our knowledge, this is the first observation of such a correlation, demonstrating how a large set of active-site variants can yield insights on sequence-structure-activity relationships even in a well-studied protein. Moreover, the designs we sorted specifically for spectral shifts indeed displayed significant shifts in excitation spectra (Fig. 5B, and Supplementary Fig. 14).

We examined the design models for a molecular explanation of the large observed differences in stability and quantum yield. For instance, Tyr145Phe, seen previously to enhance stability and quantum yield[47], appeared in all five high stability/brightness designs but only in one of the bottom 26 designs. Similarly, Thr203His, likely to stabilize the chromophore through π-π stacking interactions[48], is seen in all top designs and none of the bottom ones. Ser205Thr is in three of the top-five designs and none of the bottom. By contrast to the two mutations above, Ser205Thr is enriched in designs with high thermostability and quantum yield though we are unaware of previous studies that pointed to its significance.

We also observed large variability in photostability, which is the resistance of the chromophore to bleaching by bright light. Bleaching is often a limitation in long-term live-imaging studies[49], whereas it is an advantage in assays such as fluorescence recovery after photobleaching (FRAP), in which fast fluorescence decay enhances signal[50]. We isolated two designs that exhibited higher photostability than GFP (photostable.1 & photostable.2, with seven mutations each from PROSS-eGFP) and many significantly less photostable designs (Fig. 5A

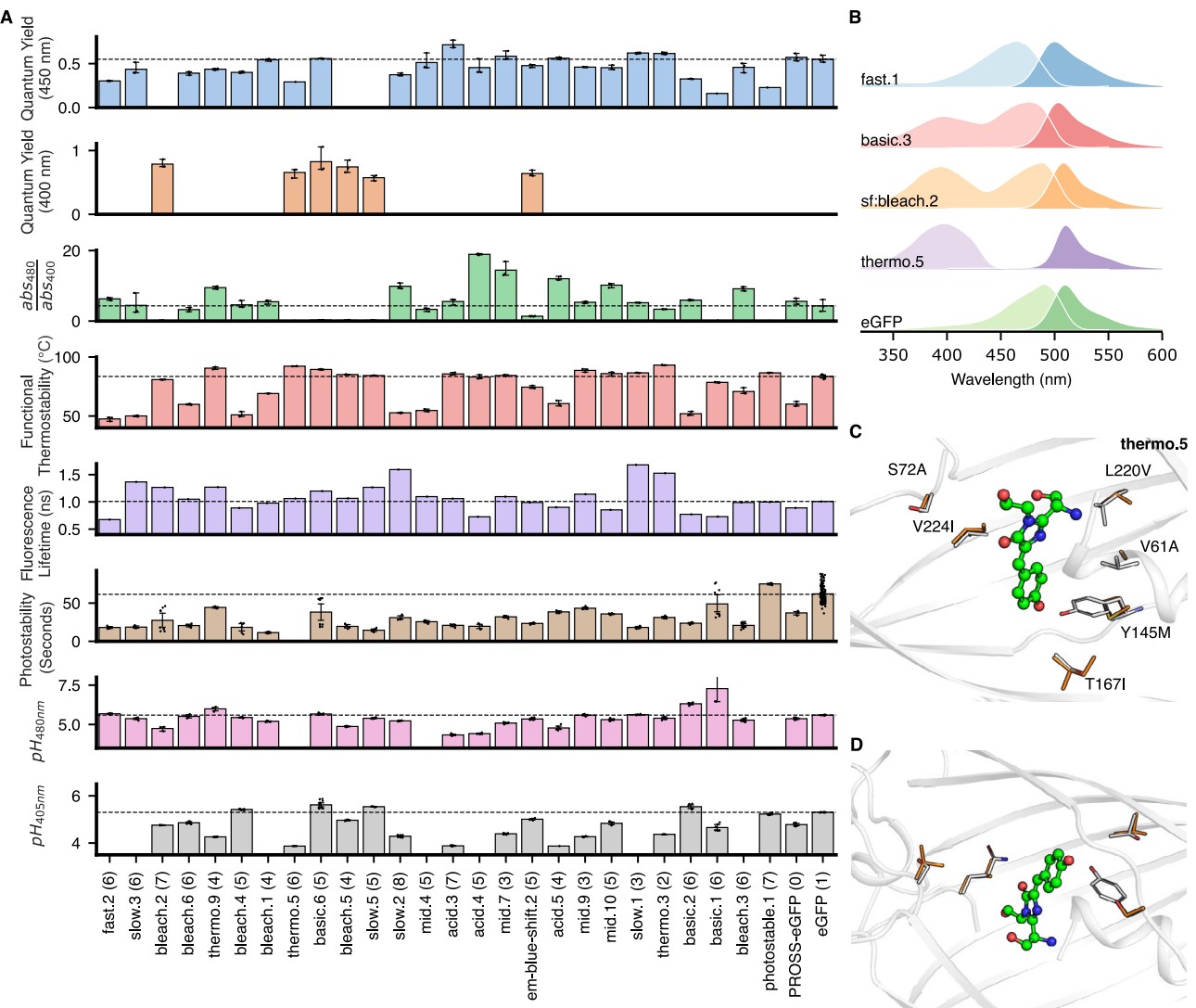

**Fig. 5 | Functional diversity among htFuncLib designs. A** A subset of tested designs clustered by sequence similarity. The dashed line marks eGFP, data are presented as mean values ± standard deviations. $n = 3$ biologically independent samples were used for quantum yield, thermostability, photostability and spectral measurements. One experimental sample was used for fluorescence lifetime measurements. **B** Selected excitation and emission spectra in light and dark hues, respectively. The excitation spectra of several designs are considerably different from eGFP (Supplementary Fig. 14). **C, D** Structural view of thermo.5 and slow.3. Each design exhibits six mutations from PROSS-eGFP. PROSS-eGFP and designs colored gray and orange, respectively. Data are provided as a Source Data file.

and Supplementary Fig. 15). At the extremes, design fast.4 (6 mutations) photobleaches tenfold faster than GFP, while the design photostable.1 requires 122% of that time. Finally, we also noted large differences in fluorescence lifetime (Supplementary Fig. 16) and pH sensitivity (Supplementary Fig. 17). Furthermore, several mutations enriched in designs with altered $pK_a$ are either adjacent to His amino acids or introduce a novel His (Thr203His). Interestingly, seven designs exhibit different pH sensitivity profiles when excited at either 405 or 488 nm (Supplementary Fig. 17).

## Discussion

Epistasis is a significant constraint on the emergence of new activities in proteins and other biomolecules[15]. Until now, experimental methods to address epistasis have relied on iterative cycles of diversification and selection, but such processes do not efficiently cover the space of functional variants. Computational methods have used evolutionary couplings among pairs of positions[51], but such analyses require deep and diverse sequence alignments, which are not generally available. Other approaches have used machine-learning models trained on high-quality and large-scale mutational data to recommend

mutations[37,52,53]. By contrast, htFuncLib only requires a molecular structure (or model) and a limited sequence alignment of homologs. Its success in generating an order of magnitude more functional active-site mutants than were previously known for GFP verifies our underlying assumption that energetically incompatible mutations are a significant source of epistasis. Furthermore, because the designs are diverse and only target the chromophore-binding pocket, they exhibit potentially useful functional diversity in each of the properties we assayed.

Our implementation of htFuncLib did not target a specific functional outcome, except for protein stability and compatibility with the chromophore. This implementation is especially suitable if multiple different and potentially incompatible functional properties are desired. For example, FRAP experiments require fluorescent proteins that bleach quickly, whereas long-term imaging experiments require slow bleaching, and we recovered designs that exhibited both properties from a single library. If a specific functional goal is desired and the molecular underpinnings of that goal are known, they can be imposed during the design process. The high stability and brightness of the eGFP starting point are likely to be key to obtaining so many

functional variants[3,54]. Further research is needed to determine whether the combination of PROSS stability design[55] and htFuncLib can access such large spaces of functional variants in less robust starting points. Previous applications of FuncLib to enzyme active sites observed function in 30-90% of the experimentally tested designs[24,56–59]. As expected, the application of htFuncLib to very large libraries exhibits a lower hit rate but generates orders of magnitude more functional designs. Therefore, FuncLib and htFuncLib are complementary approaches: FuncLib is relevant to experimental systems that are only amenable to low-throughput screening, as is the case for many enzymes, and htFuncLib to ones that can be tested at medium-to-high throughput, such as binders and fluorescent proteins.

In a companion paper, we demonstrate that the EpiNNet strategy is general and can be extended to design large and highly functional enzyme libraries comprising substantial backbone conformational diversity, including insertions and deletions[36]. We envision that htFuncLib will provide a platform for designing high-yield multipoint mutation libraries in a range of applications, including optimizing binding affinity[60] and enzyme catalytic rate and selectivity.

## Methods

### Library design

**Phylogenetic analysis.** A phylogenetic analysis was conducted as previously[24,55,61] using all sequences in the lineage of avGFP according to FPBase[35]. A total of 153 sequences were retrieved from FPBase, all synthetic variants of avGFP. Briefly, sequences were clustered by cd-hit[62] and aligned using MUSCLE[63]. The resulting multiple sequence alignment (MSA) was segmented by secondary structure elements. A position-specific scoring matrix (PSSM)[64] was then derived from the MSA segments and concatenated to create a PSSM for the whole sequence. The PSSM is used to filter mutations absent in the PSSM at each position and to bias the Rosetta energy function towards mutations favored by the PSSM (high PSSM score).

**Refinement and mutational scan.** PROSS-eGFP was modeled based on a high-resolution X-ray structure of eGFP (PDB code: 2WUR [https://doi.org/10.2210/pdb2wur/pdb]). The eGFP-PROSS model was subsequently refined in Rosetta as described before[55]. Chromophore pocket positions were then manually selected, 14, 16, 18, 42, 44, 46, 61, 64, 66, 68, 69, 72, 108, 110, 112, 119, 123, 145, 150, 163, 165, 167, 181, 185, 201, 220, 224 and 42, 44, 61, 62, 69, 92, 94, 96, 112, 121, 145, 148, 150, 163, 165, 167, 181, 183, 185, 203, 205, 220, 222, 224 for the hbonds and nohbonds libraries, respectively. All positions are within 8 Å from the chromophore, and their side chains are buried within the GFP β-barrel. All mutations with PSSM scores >−2 were then scanned in silico, as previously described[24,55]. Briefly, each mutation is modeled, refined, and scored separately on the PROSS-eGFP background. This step calculates the $\Delta\Delta G$ between the mutant and eGFP-PROSS.

**Spatial partitioning and sequence space selection.** We split the chromophore pocket into spatial neighborhoods, with each selected position as a center of a distinct neighborhood. In order to capture direct epistatic interactions, each neighborhood is extended to all positions that interact directly with the neighborhood's center. Here, direct interaction is defined as having at least two heavy atoms within 6 Å of the neighborhood's central residue. Neighborhoods were manually examined, and positions that did not interact directly with the neighborhood's center were removed. By selecting neighborhoods this way, we ensure overlap between proximal neighborhoods. These overlaps ensure that no position–position interactions are missed.

**Partial modeling and scoring.** The number of designs encoded in each neighborhood is calculated for each $\triangle G_{mut-wt}$ threshold. The energy threshold is selected to limit the number of unique variants to

under 1 million. In this particular case, the $\Delta\Delta G$ thresholds were set to +5.5 and +6.0 Rosetta energy units (R.e.u.) for nohbonds and hbonds, respectively. Neighborhoods with a sequence space smaller than 10,000 designs were fully modeled. For larger neighborhoods, only 10% of the sequence space was modeled. RosettaScripts[65] and command-line arguments for modeling calculations are in the Supplementary Information.

**Data aggregation and EpiNNet training.** We train an EpiNNet neural net model to predict which designs are more stable than PROSS-eGFP. Specifically, designs that score better than the wild-type are labeled as success (1), and the worse 50% are labeled as failed (0). Intermediate designs are considered undetermined and discarded from subsequent analysis. The resulting data are split into a training (80%) and a test (20%) set. We then train a multi-layer perceptron classifier with a single hidden layer the size of the number of selected positions. The classifier is trained on a one-hot encoded representation of the sequence data to classify whether a sequence is more or less stable than PROSS-eGFP. The classifier is trained up to 2000 iterations. Next, we rank single-point mutations according to the trained model: each single-point mutation in the tolerated sequence space is fed into EpiNNet separately and its score is recorded. The mutations are then ranked from top to bottom according to their scores. Mutations are selected for the library by iteratively adding the top-ranked mutations until the resulting sequence space reaches the experimental limit of several million sequences.

**In silico testing of the enriched versus the original sequence spaces.** To ensure the resulting sequence space is enriched for low-energy sequences, 10,000 random sequences from both the original and enriched sequence spaces were modeled and scored (using the same protocol as in the modeling step). The resulting score distributions were compared (Figs. 1D and 2A).

**Random forest analysis.** To augment the sequence data for machine-learning prediction, we added several features based solely on the sequence and not requiring atomistic calculations. These include the amino acid identity at each variable position, the total number of mutations compared to PROSS-eGFP, the number of mutations at each spatial neighborhood, and the number of mutations in specific areas. In addition, for every variable position, the difference in the surface accessible solvent area (SASA), PSSM score, and amino acid category were also assigned (comparing the mutated amino acid and the PROSS-eGFP identity). The mean and max values of each of these parameters were added as well. Non-informative features and features with low importance in initial random forest training were removed. A prediction pipeline with two consecutive elements was trained. The first predictor classifies sequences as either functional or non-functional. The subsequent predictor classifies all functional sequences as either GFP, AmCyan, GFP/AmCyan, or non-functional. Both models are gradient-boosting random forests from the LightGBM library[66].

**Visualization methods.** Visualization method as previously described[42]. Briefly, we construct a model of molecular evolution where a population evolves via single amino acid substitutions, and the rate at which each possible substitution becomes fixed in the population reflects its selective advantage or disadvantage relative to the currently fixed sequence. More specifically, in our model, the rate of evolution from sequence $i$ to any mutationally adjacent sequence $j$ is given by

$$Q_{ij} = \frac{S_{ij}}{1 - e^{S_{ij}}}$$

where $S_{ij}$ is the scaled selection coefficient (population size times the selection coefficient of $j$ relative to $i$), time is measured relative to the amino acid mutation rate (each possible amino acid mutation occurs at rate 1), and the total leaving rate from each sequence $i$ is given by $Q_{ii} = -\sum_{j \neq i} Q_{ij}$. In the current context, sequences are either predicted to be fluorescent or not, and so we set $S_{ij} = c$ if $j$ is fluorescent and $i$ is not, $S_{ij} = -c$ if $i$ is fluorescent and $j$ is not, and otherwise $S_{ij} = 0$ so that $Q_{ij} = 1$, corresponding to neutral evolution. For this analysis, we choose $c$ so that in the long-term, a population spends 60% of its time at functional sequences, representing roughly a 60-fold enrichment of functional sequences due to natural selection.

Given the rate matrix $Q$ for our evolutionary model, we then construct the visualization by using the subdominant right eigenvectors associated with the smallest magnitude non-zero eigenvalues of this rate matrix as coordinates. This produces a visualization that reflects the long-term barriers to diffusion in sequence space, and, in particular, clusters of sequences in the visualization correspond to sets of initial states from which the evolutionary model approaches its stationary distribution in the same manner, and multi-peaked fitness landscapes appear as broadly separated clusters with one peak in each cluster. Moreover, by scaling the axes appropriately, as is done here, these axes can be given units of sqrt(time), and it can be shown that the resulting distances reflect evolutionary times under this model. In particular, using these coordinates, the squared Euclidean distance between arbitrary sequences $i$ and $j$ optimally approximates (in a specific sense) the sum of the expected time to evolve from $i$ to $j$ and the expected time to evolve from $j$ to $i$. See ref. 42 for details. Calculations and plots were performed using gpmap-tools python library (https://gpmap-tools.readthedocs.io/en/latest/).

**Logistic regression and sequence logos.** L2-penalized logistic regression models were fit using scikit-learn[67]. Specifically, the global model using all sequences was fit using non-penalized regression, while the model in the neighborhood of the alternative functional sequences highlighted in Fig. 4D was fit using L2-penalization under one-hot encoding, using tenfold cross-validation to optimize the hyperparameter controlling the strength of the regularization. The regularization constant was chosen to be $C = 0.5$ as the strongest regularization before a drop in the cross-validated AUROC. Sequence logos were plotted using logomaker[68].

## Experimental procedures

**Library cloning.** Each designed library was cloned separately using a Golden-Gate assembly[38]. A computational algorithm optimizes a set of Golden-Gate overhangs to minimize the total cost of ordered oligos required to cover all mutations in the library without introducing unwanted mutations. This results in several variable and constant segments, with and without mutations (Supplementary Dataset 8). Constant segments were PCR amplified with primers adding *BsaI* recognition sites. These and all other DNA fragments were purified using (HiYield Gel/PCR DNA Fragments Extraction Kit, Real Genomics). Variable segments were ordered as degenerate oligos (Single-stranded IDT, DNA). The single-stranded oligos were double-stranded by a short PCR with a single primer and purified (Roche, KAPA HiFi HotStart ReadyMix). The concentration of each segment was measured using NanoDrop One (Thermo Scientific). A Golden-Gate assembly was conducted using the manufacturer's specifications. Briefly, all segments are added at an equal amount, without the vector, and assembled using T4-ligase and BsaI-HF-v2 using cycles of 16°C and 37°C (New-England Biolabs, M202 and R3733, respectively). The resulting assembly is PCR amplified (KAPA HiFi HotStart ReadyMix) to add the final gates and assembled into a pBAD vector with appropriate gates.

**FACS sorting.** *E. coli* BL21 (DE3) (E. cloni EXPRESS BL21 (DE3), Lucigen, #60300-1,) cells were transformed with the pBAD plasmids containing

the libraries and grown overnight. Transformation efficiency was estimated by plating serial dilutions of the transformed bacteria, ensuring that, for each library, the number of transformed cells was at least tenfold higher than the designed library size. 1 µl from each transformation was plated in dilution to estimate transformation efficiency. Cells were diluted 1:200 in 2YT media (Sigma Aldrich, #Y2377), grown to 0.6–8OD, induced using 0.2% arabinose (Sigma Aldrich, #A3256), and shaken at 20 °C overnight. Induced cultures were transferred to 4 °C for another night to allow maturation. Cells were centrifuged at $1400 \times g$ for 10 min, decanted, and resuspended with cold PBS (Sartorius #02-020-1A) twice. The cells were then sorted using a FACS AriaIII (BD Biosciences) with a 70 µm diameter nozzle and a cell flow rate of 10,000–20,000 events per second. A preliminary sorting gate was done on forward scatter (FSC) Vs. side scatter (SSC) parameters to select single bacteria cells alongside the AlexaFluor488 (excitation at 488 nm, emission detection at 530 ± 15 nm) and AmCyan (excitation at 405 nm, emission detection at 525 ± 25 nm) channels. Sorted cells were collected in SOC media (Thermo Fisher, #15544034, Thermo Fisher), grown overnight at 37 °C and transferred to 2YT supplemented with ampicillin. Plasmids were extracted by mini-prep (Qiagen, #27104). Plasmids from sorted populations were extracted by mini-prep, transformed and sorted again (using the same procedure) to reduce false-positives.

**Deep sequencing.** Plasmids from presorted and sorted populations were PCR amplified (KAPA HiFi HotStart ReadyMix) using primers to generate 590 bp amplicons, containing all variable positions excluding position 16 (forward primer: GGGCGATGCCACCTACGGCAAG and reverse primer: GAGTGATCCCGGCGGCCTC). Amplicon libraries were prepared at the Weizmann Institute's Israel National Center for Personalized Medicine. Libraries were prepared from 20 ng of DNA, as previously described[69]. Libraries were quantified by Qubit (Thermo fisher scientific) and TapeStation (Agilent). Sequencing was done on a Miseq instrument (Illumina) using a V3 600 cycles kit (Illumina, # MS-102-3003), using paired-end sequencing. Sequences were analyzed using the LAST software package and python[70,71]. Fastq sequences were aligned to all designed oligos using the LAST align function. Sequences were consequently filtered for low LAST scores, and assigned to the best aligned oligo. Pair-end reads were identified using MiSeq UMIs (unique molecular identifiers). Enrichment values were calculated as the ratio between read frequencies in the sorted and appropriate unsorted samples. The presorted libraries are too large to be completely covered by the deep-sequencing analysis. We, therefore, did not expect to detect all combinations, specifically in the nohbonds library. However, given that the transformation efficiency was greater than $5 \times 10^7$, and $>10^8$ cells were sorted by FACS, it is likely that the majority of the functional designs were recovered. We thus considered all sequences found solely in the sorted samples to be enriched. The sorted library will be deposited in AddGene upon publication.

**Cloning of single designs.** Genes encoding for selected designs were ordered from Twist Bioscience and codon-optimized for *E. coli*. Genes were inserted in the pET28 vector using *BsaI* restriction sites previously cloned using QuickChange. All plasmids were sequence verified. Designs selected directly from FACS sorting were transferred from the pBAD vector into pET28 by PCR amplifying (KAPA HiFi HotStart ReadyMix) the insert with primers and adding *BsaI* recognition sites. Amplicons were purified and inserted into a pET28 vector with *BsaI* insertion sites using Golden-Gate assembly (New-England Biolabs, #M202 and #R3733, respectively). All plasmids were individually verified using Sanger sequencing.

**Protein expression and purification.** pET28 plasmids containing the relevant insert were transformed into BL21 (DE3) (Thermo Fisher, #EC0114) cells and grown overnight. Overnight cultures were diluted

1:100 in 10 ml conical tubes containing 2YT and 50ug/ml kanamycin (Sigma Aldrich, #BP861), grown to OD = 0.6–8, induced using 1 mM IPTG (Sigma Aldrich, #I6758), and shaken overnight at 20 °C. After expression, samples were shaken at 4 °C to maximize chromophore maturation. Samples were centrifuged at 2480 × g for 20 min at 4 °C and resuspended in 1 ml lysis buffer containing PBS, 0.01% Triton x100, 0.02% Benzonase, 0.1% protease inhibitor cocktail, and 0.1 mg/ml lysozyme (Sigma Aldrich, #9036-19-5, #E1014, #P2714, and #MAK405). Samples were then sonicated and centrifuged at 18,400 × g at 4 °C for 45 min. 500 μl Ni-NTA beads (EMD Millipore, #LSKMAGH10) per sample were resuspended in PBS and allocated into an appropriate number of 1.7 ml tubes. The supernatant of each sample was transferred to a tube containing 500 μl Ni-NTA beads and 10 mM imidazole (Sigma Aldrich, #288-32-4). Samples were shaken at room temperature for 2 h for binding, centrifuged at 1400 × g for 3 min, and the supernatant was removed. Beads were resuspended in PBS with 20 mM imidazole and shaken for 30 min at room temperature. Samples were centrifuged again at 1400 × g for 3 min, and the supernatant was removed. Samples were eluted using PBS with 500 mM imidazole, shaken for 1 h at room temperature, and centrifuged at 1400 × g for 5 min. The supernatant was recovered and kept at 4 °C. Protein purity was estimated by SDS-PAGE gel electrophoresis, and protein concentration was determined using NanoDrop One (Thermo Scientific).

**Functional thermostability.** Functional thermostability was measured similarly to SYPRO orange measurements[72]. 10 μM of each design were diluted in PBS in triplicates and placed in a 96-well plate (20 MicroAmp Fast Optical 96W Reaction Plate, Thermo Fisher, and MicroAmp Optical Adhesive Film). A ViiA7 real-time PCR instrument (Applied Biosystems) was used to measure fluorescence during heating from 25–99.9 °C at 0.05 °C/s. Raw data were analyzed using Python to find the temperature at which fluorescence was 50% of the max for each well.

**Fluorescence lifetime.** Fluorescence lifetime measurements were performed using a MicroTime200 optical setup. GFP samples were placed as drops on top of 175 μm glass slides (Precision Cover Glass No:1.5H, Marienfeld), mounted on an inverted microscope (IX83 inverted, Olympus) with a 60× water immersion objective (UPlanSApo, Superapochromat, Olympus). A 485 nm pulsed-interleaved excitation laser (LDH-D-C-485, PicoQuant) with a repetition rate of 20 MHz (50 ns) was directed via a dichroic mirror (ZT473/594rpc, Chroma) and focused -10 μm into the sample. The fluorescence emission signal passed through a 50 μm pinhole and an emission filter (HC520/35, Semrock). Photons were focused into a single-photon avalanche diode (SPCM-AQRH-14-TR, Excelitas) coupled to a counting module (Pico-Harp 400, PicoQuant), and time-correlated single-photon counting (TCSPC) histograms were generated. Each sample was measured for 1–5 min with laser intensities between 2–20 μW, adjusted using OD filters to reach a photon count rate of -20 kHz. The profile for the instrument response function (IRF) was obtained by measuring scattered light from a mirror. The fluorescence decay curves were fitted with a bi-exponential fluorescence decay model by iterative IRF-reconvolution to extract the characteristic lifetimes and weights of the GFP designs.

**Photobleaching.** Photobleaching was measured similarly as previously described[73]. A final concentration of 1 μM of each variant was embedded in polyacrylamide gels (168 μl 30% acrylamide/bis-acrylamide, 25 μl PBS, 0.5 μl TEMED, and 3 μl 10% APS (Sigma Aldrich, #A3574, #110-18-9 and #SML2389, respectively) and 57 μl fluorescent protein in PBS) inside microscope slides (ibidi, μ-Slide 8-well). Slides were mounted to Eclipse TI-E Nikon inverted microscope (Nikon Instruments Inc., Melville, NY) with Plan Apo DIC 60X/1.4 NA objective and equipped with a cooled electron-multiplying charge-coupled device

camera (IXON ULTRA 888; Andor). The measurement consisted of six repetitions of exposure to the strongest available LED light at either 405 or 488 nm for 15 min while capturing an image every five seconds. Images were analyzed using ImageJ to recover the mean intensity from each frame. A bi-exponential function was fitted to normalized brightness as a function of exposure time. The weighted average of the exponential coefficients was calculated. Outliers were removed, and at least three measurements were used to calculate means and standard deviations.

**Fluorescence spectra and quantum yield.** Proteins were diluted in PBS to OD 0.05 at either 450 or 400 nm in disposable fluorescence cuvettes (ordered from Alex Red No CUV010015) in triplicates. OD was measured on a Cary 60 UV–Vis spectrophotometer (Agilent Technologies) from 300 to 650 nm. Both emission and excitation spectra were measured with the same samples on a fluorescence spectrophotometer (Varian Cary Eclipse). Quantum yield was calculated using the relative method described in the literature[73,74]. Briefly, the ratio between absorbance at the excitation wavelength and the integral of emission spectra are measured for each sample and a standard with known quantum yield. Fluorescein and 1-aminoanthracene were used as standards for measurements at excitation wavelengths 450 and 400 nm, respectively.

**pH sensitivity.** Buffers at pH ranging from 3.0 to 10 were prepared as previously reported[73]. One hundred microliters of pH buffer were placed in black flat-bottom 96-well plates (Greiner Bio-One, #655090), and 2 μg of fluorescent proteins were added. Samples were incubated at room temperature for 1 h, and fluorescence at both 405 and 488mn and emission at 520 nm was measured for all wells (Tecan, Infinite M Plex).

**Multipoint mutants from other GFP datasets.** To compare the GFP variants considered here with that studied earlier, we extracted from previous works (refs. [17,18]) sequences and fluorescences of variants having mutations in the chromophore pocket positions only (corresponding to 2WUR GFP positions 14, 16, 18, 42, 44, 46, 61, 62, 63, 64, 66, 68, 69, 72, 92, 94, 96, 108, 110, 112, 119, 121, 123, 145, 148, 150, 163, 165, 167, 181, 183, 185, 201, 203, 205, 220, 222, 224). The four GFP variants are *Aequorea victoria* GFP (avGFP[4]), *Aequorea macrodactyla* GFP (amacGFP[3]), *Clytia gregaria* (cgreGFP[3]), and *Pontellina plumata* GFP (ppluGFP[3]). Our reference GFP sequence was aligned to the alignment of the avGFP, amacGFP, cgreGFP, and ppluGFP.

**Fluorescence versus the number of mutations in the active site.** The fluorescence F of different GFP variants was fit to the exponential-decline and negative-epistasis function[40]:

$$F = \exp(-\alpha n - \beta n^2), \qquad (1)$$

Where *n* is the number of mutations. Equation (1) can be rewritten as:

$$F = \exp\left(-An - B\frac{n(n-1)}{2}\right), \qquad (2)$$

Where $A = \alpha - \beta$ represents robustness, and $B = 2\beta$ represents epistasis. We required $A \geq O$ and $B \geq O$. The analysis is provided as a Jupyter notebook 'GFP_threshold_epistasis.ipynb'.

**Statistics and reproducibility.** Quantum yield and all spectral measurements were conducted with $n = 3$ biological samples. Functional thermostability was measured with $n = 3$ biological repeats on separate dates and plates. Fluorescence lifetime measurements were conducted once for all samples except for eGFP, which was measured three times. Measurements were conducted until >10,000 photons were detected

at the peak; Fluorescence lifetime of eGFP was measured with $n = 3$ biological samples, with mean $2.2 \pm 0.1$ (ns), indicating high reproducibility. Photostability was measured with $n = 6$ biological repeats, except for eGFP which was measured in all tested slides; we excluded bleaching curves for which we could not fit our equations with $r^2 < 0.9$.

### Reporting summary

Further information on research design is available in the Nature Portfolio Reporting Summary linked to this article.

## Data availability

All sorted and presorted libraries, together with designs of special interest, were deposited to AddGene (deposit number 81660). The deep-sequencing data generated in this study have been deposited in the Figshare database under accession code 21922365 [https://doi.org/10.6084/m9.figshare.21922365]. Source data are provided with this paper.

## Code availability

Jupyter notebooks for the evolutionary analysis can be found at https://bitbucket.org/cmartiga/gfp_core/src/master. Jupyter notebooks, including data, for other analyses and the htFuncLib algorithm, are available at https://github.com/Fleishman-Lab/htFuncLib[75].

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

## Acknowledgements

We thank Moshe Goldsmith, Robert Campbell, and Yi Shen for helpful advice and Olga Khersonsky and Shiran Barber-Zucker for critical reading. Research in the Fleishman lab was supported by the Volkswagen Foundation (94747), the Israel Science Foundation (1844), the European Research Council through a Consolidator Award (815379), the Dr. Barry Sherman Institute for Medicinal Chemistry, and a donation in memory of Sam Switzer. Research in the McCandlish lab was supported by NIH grant R35GM133613, an Alfred P. Sloan Research Fellowship, and additional funding from the Simons Center for Quantitative Biology at Cold Spring Harbor Laboratory.

## Author contributions

J.Y.W. designed and performed the design computational and experimental work and wrote the paper; C.M.G. performed the fitness landscape analysis and wrote the paper; R.L.S. developed the EpiNNet algorithm; S.Y.H. assisted in algorithm development; D.L. designed and performed experimental work; R.N. designed and performed experimental work; H.W. provided experimental assistance and resources; E.P.K. performed experimental work; D.M. provided guidance in analyzing GFP functional diversity; D.I. performed data analysis; D.M.M. supervised the fitness landscape analysis and wrote the paper; S.J.F. supervised the algorithm and experimental work and wrote the paper.

## Competing interests

The authors declare no competing interests.
