## [Peer Review File · Nature Communications]

Designed active-site library reveals thousands of functional GFP variantsReviewer #1 (Remarks to the Author):

The paper by Weinstein et al describes a computational method called htFuncLib to generate functional variants, and then an experimental testing and evaluation of the method using GFP. The method and results appear promising and potentially very useful for generating libraries for large-scale testing, and could thus be useful in other engineering efforts. In particular, by enabling efficient combinatorial engineering in for example active sites, the method could be used to optimize for example enzymes.

The paper is relatively dense, with several novel ideas and results. Thus, while each part of the paper is well written, it is also easy as a reader to get lost because there are many points made, and because many of them are presented only at an overall level. I missed a lot of details of the results and methods, which made it difficult to evaluate the work. The comments below should also be seen in this light; there may be points that I have missed, but in that case this could be because the points/results/methods are in places very difficult to find.

Briefly described (if I understand the method correctly), the application to GFP starts with manually selecting 27 sites around the chromophore assumed to be important for function and followed by these steps: A first filtering is based on a phylogenetic analysis and single-site Rosetta calculations and results in a list of promising amino acid substitutions at a subset of the 27 sites. Then, Rosetta calculations of multipoint mutations constructed in spatially local neighbourhoods are used as input for a neural network model that outputs a ranked list of substitutions from which a selected top is used to construct a combinatorial library. Two such libraries were made and screened for fluorescence in high throughput and for two excitation wavelengths. Screened variants from the two libraries were pooled and a common random forest model was trained to predict the functional state of all 11 plus 0.93 million combinatorial variants of the two libraries respectively. The usefulness of individual substitutions in functional multipoint mutants is analysed based on the predicted function states.

Overall, I find the paper interesting and it appears that the method can make combinatorial libraries that are enriched in successful and potentially interesting functional variants. Although it is not surprising to find interactions between buried and spatially close sites, the paper presents a method that seem capable of handling this, i.e. that a substantial fraction of the variants generated are functional. How many and what fraction is still unclear (see below for discussion on this). One area to improve the manuscript is that the computational method, details of the high-throughput screen and random forest model are relatively poorly described, and in places also poorly validated, which leaves substantial uncertainties for the reader, and makes it very difficult to evaluate the method presented.

More detailed comments and suggestions for improvements.

1.

There are some claims in the abstract that are difficult to find support for in the paper:

1a.

"We screened 11 million htFuncLib designs that targeted the GFP chromophore-binding pocket ...". The experimentally screened library in theory has 11 million members but I do not see evidence that all have been transformed and screened. Fig. 2C suggests a maximum of $\sim 10^5$ experimentally screened variants. If the abstract refers to the in silico screen using the random forest model, please make that clear, and also say how many are known to have been screened experimentally.

1b.

"... and isolated >16,000 unique fluorescent designs encoding as many as eight active-site mutations". In the body of the paper the 16,000 functional variants are referred to as "putative" or "potentially active" which I find more reasonable due to the "Relatively loose criteria (enrichment in the selected population relative to the presorted population >1)" by which these are identified. If the authors wish to claim to have identified 16,000 functional variants, the rest of the paper

should be more focused on validating this, see in particular below regarding the uncertainty of the high-throughput assay.

1c.

"By eliminating incompatible active-site mutations, htFuncLib generates a greater diversity of functional sequences than evolutionary or mutational scanning approaches for optimizing enzymes, binders, and other functional proteins." Fig. 3B illustrates that when only looking at variants containing functional site mutations, i.e. removing most variants from the other sets, then htFuncLib seems to cover a larger space. However, it is not clear from the abstract that the other sets are heavily reduced, in particular because variants with more mutations are likely to have at least one outside the functional site. The text in the conclusion is a bit clearer on this point, but it would be good that the abstract reflects this.

2)

Major parts of the methods are not described in any substantial detail. For example the deep sequencing and random-forest model which are described with six and three sentences respectively. This makes it almost impossible to judge the results and conclusions of the paper, and for others to build on this work. Ideally, one should be able to examine all major parts from reading the paper, for example, only details should be looked up in the code.

3)

I find it surprising that PROSS-eGFP performs relatively poorly in the computational and experimental stability evaluation (Fig. 5a). If I understand correctly, PROSS-eGFP should be optimized by something very similar to the filtering method so why does so many beneficial substitutions show up in the filtering? The only information I can find regarding the "filtering" is that this leaves a combinatorial space of 10^{18} variants; it would be interesting to see the resulting substitutions presented as in supplementary table S1. Also, Y145F is described in ref. 33 as a part of PROSS-eGFP but here reintroduced as a new mutation? Similarly, T167I is presented as a new substitution here, but seems to already be in the background in ref. 33 (looking at the DNA sequence below the supplementary table S2). Also, in ref. 33 PROSS-eGFP is reported to have 11 or 12 substitutions relative to eGFP but here only one (interpreting undescribed tics in Fig. 5A and the caption of Fig. 3: "PROSS-eGFP (and eGFP, which are nearly identical in the designed positions)". All in all, this is somewhat unclear from reading the paper. Maybe it is just me having a hard time finding the data, but then that might also be the case for other readers. It would have been useful to report exact sequences of eGFP and PROSS-eGFP (and sfGFP) and substitutions used. Also, please comment on the apparent low stability of PROSS-eGFP and the relationship to the fact that it is optimized.

4)

Regarding the number of tested designs, I cannot make the numbers in supplementary table S5 match the attached csv file: The csv holds 72 sequences incl. 3 controls, i.e. 69 tested designs that all seems to be functional, whereas the text reports 68 designs tested in total? There may be a way to add this up but I could not find it. The 19 variants in sfGFP background are listed in the csv but not in table S5. Also, I can only see 15 functional random-forest designs and 10 AmCyan variants selected for spectral shift in the csv but 16 and 14 are reported in table S5? Most importantly, please report the sequences and number of mutations of the failed design (e.g. in the csv and/or in the text) as this is essential information for the community. Also, please describe in the paper the nine "Additional designs" listed in Table S5 as selected from deep sequencing, of which only one is functional. Again, apologies if these were presented somewhere, but I could not find it.

5)

In the top plot of Fig 3A, it seems that 100% of single variants are functional – perhaps this is WT, i.e. zero mutations, and the x-axis is shifted? If this is the case, the RF model seems very optimistic in identifying function with almost 100% single mutants functional (currently $x=2$) or at least more than NGS. With this, I find it odd that the bottom plot shows slightly more functional single variants in NGS than in RF (assuming most single mutants are observed in NGS). Please check and clarify.

6)

Most striking in Fig 3A is the high fraction of the "16,000 potentially active designs" with more than four mutations. This should be validated better if authors wish to report 16,000 functional variants identified in the abstract. First, the authors only report tests of designs up to 8 mutations whereas a substantial fraction of functional NGS variants have more than 8 mutations. Please report the success rate in experimental tests per number of mutations (also related to pt. 4 above on information about the failed designs). Second, please comment on the statistics in supplementary Table S2: The false-positive rate is 1/12, i.e. ~8% are the falsely predicted functional out of the actual non-functional. This is a quite high number since the paper reports ~90% non-functional (Fig. 2C), i.e. with 100,000 non-functional variants, 8,000 are expected to be false positive. This reflects the imbalance in training on a high-prevalence set (most functional) and applying to a low-prevalence set (fewest functional). Third, there is very little indication of the uncertainty in the NGS experiment. It requires a very deep sequencing to cover 100,000 unique variants without paired-end reads, to an extent that warrants calculation of enrichments. To calculate enrichments, the authors need to have a good idea about the abundance of a variant prior to selection and there is no discussion on how this is addressed. Please comment on this, e.g. size of transformed library, how many cells are expected to have more than one plasmid, FACS coverage (cells sorted per library size), sequencing coverage (average number of reads per unique variant), which region of GFP is sequenced (maximum 600 bases are sequenced), are all functional variants observed in the non-sorted sequencing, are pseudo-counts applied, frequency of unexpected substitutions and how these are handled, etc.

7)

It would be interesting to see some more details on the construction of the neighbourhoods. E.g. a supplementary table listing the sites of the filtered mutations could also list the neighbourhood of each site and the number of calculated multipoint mutations. Are these mostly double mutants or higher order mutants?

8)

It would be appropriate to make some quantitative comparison with the previous version of FuncLib (ref. 24), e.g. by the success rates obtained in experimental validations.

9)

It would be interesting to know with a bit more detail on the phylogenetic analysis. The authors write "In this selection step, we keep mutations that are likely to be present in the natural diversity of sequence homologs and that are moreover predicted not to destabilize the protein native state according to atomistic design calculations³⁵". GFP is sometimes considered not to have very many natural homologs and fpdb.org (ref 35) contains a lot of synthetic variants. Please give the number of sequences in the phylogenetic analysis and, if possible, indicate how many of these are natural, e.g. belongs to a reference genome.

10)

The methods section describes "An alternative mutation selection approach that uses Integer Linear Programming" which is only briefly referenced in the text. This should either be removed or the authors should show the results.

11)

In the paragraph starting with "Our working hypothesis is that epistatic interactions most frequently arise from three molecular sources (Supplementary Figure 1)" the third point is unclear and not illustrated in supplementary Fig. S1: "(3) stability-mediated interactions caused by the nonlinear relationship between the free energy of folding and the fraction of natively folded and functional protein".

Minor points

1)

It would be helpful if Fig. 1 more directly illustrated what "filtering" and "EpiNNet enrichment" means and where in the pipeline it is performed

2)

Should T65S be in supplementary Fig. S2? It would be useful for the discussion in Fig. 4

3)

Fig. 4A caption "GFP488/53" should be "GFP488/530"

4)

In methods section under FACS sorting: "E. cloni" should probably be "E. coli", though I quite like the name "cloni"

5)

In the introduction the authors write "and functional multipoint mutants are exceptionally rare", but do not provide a reference to this general statement. Similarly with the statement "Epistasis is a key reason for the low tolerance to multipoint active-site mutations."

Reviewer #2 (Remarks to the Author):

In this study, Weinstein and colleagues use a combination of energetic modeling and high-throughput screening to identify GFP variants with multiple mutations, addressing the challenge of potential negative epistasis between mutations reducing the hit-rate. htFuncLib was used to design a set of point mutations and then combinations of mutations that were energetically favorable. Then, a machine learning EpiNet model was trained to discriminate favorable and unfavorable combinations of mutations. Hits from this approach included those with > 8 mutations, which exceeded the tolerated mutational perturbation load from previous design approaches. This work and a companion submission on enzyme engineering show that issues with epistasis can begin to be addressed by combining judicious energetic modeling combined with training of machine learning models. An important and relevant study to the protein engineering field. The work is technically sound and clearly presented.

Two comments that should be addressed:

(1) There is no functional goal in these libraries - i.e. quantum yield, photostability, color. How would these methods be adapted if a particular functional feature, not just structure and stability were to be optimized. Excitation and emission spectral properties were not described (peak wavelenths). Can models be trained to identify what features contribute to photophysical properties?

(2) This training of EpiNet should be discussed in the context of the choice of host protein - a stable version of GFP - and specifically the work earlier this year from Kondrashov (<https://doi.org/10.7554/eLife.75842>) showing that mutational landscapes that are flatter are not as useful for training models. Does the energetic modeling in htfuncLib work for more 'fragile' proteins where epistatic interactions can have a more pronounced effect on folding/function?

Reviewer #3 (Remarks to the Author):

In this manuscript, the authors introduce the htFuncLib, a protein-engineering pipeline to design and test variant libraries focused on protein functional sites. The motivation behind developing such a method is to increase the sampling efficiency and diversity around a protein functional site, which is usually highly conserved and sensitive to mutations. To achieve this goal, the authors have to overcome the epistatic effect by introducing multiple mutations simultaneously, which, in the past, has only been partially achieved by directed evolution through iterative mutation-selection cycles. The htFuncLib method starts with low-energy PSSM-approved mutations in the functional site (by phylogenetic analysis and Rosetta energy calculation), then ranks and selects those mutations by their mutual compatibility (by a trained neuron network EpiNet). After this in silico screening, DNA fragments encoding these compatible point mutations are assembled in an all-against-all combinatorial library by Golden Gate method, tested by high-throughput FACS, and read out by deep sequencing. The authors apply this engineering pipeline to GFP's fluorescence

functional site. The results are impressive: 1.) they explore a much bigger sequence space that is inaccessible in multiple previous attempts, 2.) the functional multipoint mutants after library selection show desired functional diversity in terms of protein stability, fluorescence spectra, fluorescence lifetime, pH sensitivity, and fluorescence photo-stability, and 3.) the molecular mechanisms of epistasis underlying the successfully selected GFP variants are interesting for structural analysis. Overall, the manuscript presents a pragmatic way to diversify certain protein functions and I anticipate it will attract attention among protein engineers working towards protein tools (eg. imaging tools such as fluorescent proteins) and enzymes, thus I recommend this manuscript for publication after a minor revision. Below are my comments for the authors:

1. A direct comparison between FunLib and htFunLib would be necessary here. If adding a perceptron-based neural network (or ILP) machine learning module largely improves the end results, it would be worthwhile to ask what Rosetta method lacks and what role Rosetta design calculation plays in this new method.
2. The general applicability for other users and other proteins is not very clear. There are several manual steps in the Method description (Line 455, 462, 470). While it is understandable to introduce manual intervention on every steps during method development and the initial application, I would like to see how the authors plan to automate the pipeline for future applications.
3. The final paragraph in the Introduction is slightly an overstatement (Line 55, "arbitrarily large libraries" and Line 61, "millions (and potentially billions) of designs"). From the Method description, it is obvious that the size of the final library is a limiting factor for designing the combinatorial library (Line 462-464, Line 494-497). I would suggest the authors to revise this paragraph to avoid misleading.
4. In Line 81-86 and Supplementary Figure 1, the authors listed three hypothesized sources for epistatic interactions. It is hinted in the text (Line 99, "penalize backbone deformation" and Line 105 "most likely to give rise to...") that the htFunLib is focused on establishing type 1 epistatic interactions only (this is my speculation). It would make the manuscript easy to understand if the authors can offer a direct correspondence between the three types of epistatic interactions and the htFunLib library design's target interactions.
5. Following comment 4, I am also confused about how the method deals with backbone movement upon introducing multiple mutations, eg. how does the calculation "penalizes backbone deformation (Line 105)"?
6. In Fig. 2D, the overlay of the top-ranked mutations could be better illustrated in a different color.
7. In Fig. 3A, "NGS" could be better named as "htFunLib-NGS". I misunderstood it as all the next-generation sequencing data combined (or, does it really mean all the data combined? See, I'm confused.).
8. In Fig. 3A, it does not make clear sense to me that the point mutants ("1" in the bottom plot) have a functional ratio of 100% ("1.0" in the top plot) for all the libraries and "RF" prediction. Is it a normalization point? If it is not a normalization point, does it indicate that the fluorescence threshold for defining "functional" is arbitrarily low in this analysis? In addition, since the other reference libraries (avGFP, cgreGFP, ...) are sorted differently, I wonder how to justify this comparison of "functional" variants.
9. Fig. 4 and Supplementary Fig. 8 are the same low-dimensional representation of protein fitness landscape labeled with "clean" Random Forest (RF) predicted functional mutations and "noisy" experimental data, respectively. The authors choose to focus their analysis and discussion on the RF-predicted results (Fig. 4) in the main text. While this is totally reasonable with proper justification (as the authors have provided in line 355-356 for "false negative" and line 885-886 for "false positive"), it should be noted if the representative mutations have strong or weak signals in the experimental data. If they are completely missed in the FACS sorting and NGS sequencing, further experimental validation is needed to support the authors' claim. For example, the discussion on the "two long parallel tails" in line 323-344 is not very convincing to me since the same signals are not apparent in Supplementary Fig. 8A. To keep this part as a novel finding, I would suggest the authors test the representative mutations experimentally.
10. Reference data of transferring mutations to sfGFP are missing (Line 361-362).
11. For the 68 unique designs (Line 349) chosen for protein purification and biochemistry characterization, how many functional-site mutations do they carry?
12. An open and honest discussion on the limitations of the htFunLib method would make this

manuscript stronger. From several places in the main text, the htFuncLib seems to require a very stable starting point and it cannot explicitly improve a specific aspect of the protein function. I think that general readers will appreciate an open discussion in this regard.

Reviewer #4 (Remarks to the Author):

The manuscript describes development and application of the htFuncLib – a computational protein design workflow combining atomistic and machine-learning based approaches. The goal of the htFuncLib is to increase efficiency of laboratory screening efforts by eliminating poorly scoring combinations of mutations from combinatorial libraries, allowing exploration of highly epistatic fitness landscapes. Even for a limited set of manually curated designable positions, an astronomically large number of combinations makes exhaustive sampling of the full sequence space computationally intractable. To optimize amino acid composition for each position the authors first used phylogenetic information and in silico site saturation mutagenesis (SSM) to identify residue types most likely individually tolerable in the parent sequence context. However, as their computations show, a dominant fraction of the designs constructed by random combination of these individually beneficial mutants has substantially worse computational score relative to parental sequence.

The authors propose a simple and elegant computational procedure that apparently helps to alleviate this problem. The entire set of designable positions is split into spatial neighborhoods and the range of allowed amino acids for each position is flexibly adjusted so as to make the total number of sequence combinations for the neighborhood computationally tractable (under 10^6). Rosetta energy score is computed for each combination (or 10% of all combinations for large neighborhoods) in each neighborhood and combinations are classified as “good” or “bad” relative to the score of the parental sequence. Authors train a neural network (EpiNNet) to classify designs, and use the trained network to rank individual mutations in a way reflecting probability of the mutation to be in a “good” scoring combination. The main discovery of the study is that designs constructed from higher ranking mutations (“EpiNNet enriched”) have a much higher chance to have better energy scores than designs constructed from a set of mutations filtered using phylogenetic information and single SSM computations.

Authors proceed to apply their workflow to construct a library of PROSS-eGFP – a previously optimized variant of avGFP. Sorting library using FACS indicated presence of the variants retaining fluorescence even with up to 8 mutations. While the functional status of the overwhelming majority of >16,000 variants was assigned based on their enrichment in NGS data, some variants were purified on a scale sufficient for more detailed biophysical analysis.

While the rationale of the method is well laid out and convincing, the findings seem a bit underwhelming.

First, it appears that library construction using high ranking mutations lead to a mostly very conservative set of allowed mutations. Considering this, the result of finding functionally active variants with up to 8 mutations in the active site becomes almost trivial. Authors remark on the inclusion of radical substitutions into the library, but it is not clear how often such mutations appear in the active variants and more importantly how often presence of the radical substitutions affect functionality in a practically significant way. Additionally, previous studies referenced in the manuscript (Somermeyer, 2022) indicated various levels of robustness for different variants of the GFP. avGFP was reported to have intermediate robustness (tolerating at most 4 mutations), but given extensive optimization of PROSS-eGFP it may be not surprising its robustness increased to the level allowing it to tolerate more mutations. It might be helpful to see how the method performs for less optimized proteins.

Second, it would be helpful to see a control experiment where a library is created by combining lowest ranking mutations, or selected positions completely randomized to be able evaluate significance of the sequence space optimization provided by EpiNNet or any other method.

Third, authors claim functional variants having a wide range of biophysical properties (spectral changes, thermostability changes, quantum yield, photostability, life time etc.). Given the multi-modality of fluorescent function and its sensitivity to the immediate environment it is not surprising to discover variants have functional diversity. It is exciting to see few variants with greatly increased thermostability or fluorescence lifetime, but on the other hand almost all variants are less photostable making their use limited to specific applications. And yet it is hard to imagine similar results cannot be obtained with other types of diversification as exemplified by multiple examples of directed evolution experiments with fluorescent proteins. Interestingly, it appears none of the characterized variants have substantial changes in emission spectra, which is quite often the most desired feature to be modified.

Fourth, it is not clear how significant is the role of EpiNNet in ranking and selection of the mutations to be allowed in the library. Alternative methods to perform this task were described and applied to optimize enzymes (Fox, 2005 DOI: 10.1016/j.jtbi.2004.11.031; Fox 2007 <https://doi.org/10.1038/nbt1286>)

On a subject of presentation.

Lines 204-205 Describe how many libraries were constructed, and were there more than just nohbonds and hbonds libraries.

Lines 213-216 Reference 20 used epPCR to construct libraries with no ability to target mutations specifically to chromophore-binding pocket, so there were probably no mutants with ≥ 5 mutations in the active site

Lines 267-269 How conservation score is defined and deltaPSSM is computed?

Lines 494-496 Not clear what role EpiNNet plays in selecting mutations for the library. It appears the EpiNNet is used to rank the mutations, but not select them. Therefore, comparison between ILP solution and EpiNNet mutation-selection process is confusing and misleading

Line 459 should it be scores greater than -2 ?

Lines 462-463 Methods section describing "Refinement and mutation scan" seems to contain text relevant for "Partial modeling and scoring".

Lines 114-115 seems to erroneously reference an ILP solution to picking ranked mutations as a method to train a model to perform classification task.

Line 584 ... and purified using <what?>

Line 618 should be pET28 instead of pBAD?

Line 619 ... restriction sites cloned introduced previously

Line 625 BL21(DE3) cells not BL21 cells? Expression from pBAD vector can be done in BL21 cells, pET vectors with T7 promoters require BL21(DE3) cells

Figure 3A top panel suggests all # mutations=1 are active in all libraries, which is probably incorrect. Also the color scheme is illegible, please make markers of different shapes to help read the graph.

Figure 1C What is the side-bar color gradient illustrating?

Page 3 line 105 and page 15 line 469 A unit of distance, presumably Å, is missing. There is also inconsistency in font on page 15.

Page 3 The definition given in parentheses of proximal positions is incomplete and misleading. Perhaps use definition given in methods which is much more effective.

Page 5 The hbond and nohbond libraries differ in their definitions only in their inclusion of positions involved in hydrogen bonds to the chromophore, but the smaller number of positions included in the hbond library indicates some additional criteria differentiating selection of positions in these two libraries

Page 52 Supplementary table 6 appears to be missing.

Page 12 line 369 Functional thermostability values are missing units, presumably °C.

We thank the Reviewers for their detailed and constructive comments. Below, please find a detailed response to all the comments. Revisions in the resubmitted manuscript are labeled using red font.

Reviewer #1 (Remarks to the Author):

The paper by Weinstein et al describes a computational method called htFuncLib to generate functional variants, and then an experimental testing and evaluation of the method using GFP. The method and results appear promising and potentially very useful for generating libraries for large-scale testing, and could thus be useful in other engineering efforts. In particular, by enabling efficient combinatorial engineering in for example active sites, the method could be used to optimize for example enzymes.

The paper is relatively dense, with several novel ideas and results. Thus, while each part of the paper is well written, it is also easy as a reader to get lost because there are many points made, and because many of them are presented only at an overall level. I missed a lot of details of the results and methods, which made it difficult to evaluate the work. The comments below should also be seen in this light; there may be points that I have missed, but in that case this could be because the points/results/methods are in places very difficult to find.

We thank the reviewer for the positive comments and for the thoughtful suggestions. In view of the comments, we simplified the presentation in several places (noted below).

Briefly described (if I understand the method correctly), the application to GFP starts with manually selecting 27 sites around the chromophore assumed to be important for function and followed by these steps: A first filtering is based on a phylogenetic analysis and single-site Rosetta calculations and results in a list of promising amino acid substitutions at a subset of the 27 sites. Then, Rosetta calculations of multipoint mutations constructed in spatially local neighbourhoods are used as input for a neural network model that outputs a ranked list of substitutions from which a selected top is used to construct a combinatorial library. Two such libraries were made and screened for fluorescence in high throughput and for two excitation wavelengths. Screened variants from the two libraries were pooled and a common random forest model was trained to predict the functional state of all 11 plus 0.93 million combinatorial variants of the two libraries respectively. The usefulness of individual substitutions in functional multipoint mutants is analysed based on the predicted function states.

Overall, I find the paper interesting and it appears that the method can make combinatorial libraries that are enriched in successful and potentially interesting functional variants. Although it is not surprising to find interactions between buried and spatially close sites, the paper presents a method that seem capable of handling this, i.e. that a substantial fraction of the variants generated are functional. How many and what fraction is still unclear (see below for discussion on this). One area to improve the manuscript is that the computational method, details of the

high-throughput screen and random forest model are relatively poorly described, and in places also poorly validated, which leaves substantial uncertainties for the reader, and makes it very difficult to evaluate the method presented.

More detailed comments and suggestions for improvements.

1. There are some claims in the abstract that are difficult to find support for in the paper:

1a.

“We screened 11 million htFuncLib designs that targeted the GFP chromophore-binding pocket ...”. The experimentally screened library in theory has 11 million members but I do not see evidence that all have been transformed and screened. Fig. 2C suggests a maximum of $\sim 10^5$ experimentally screened variants. If the abstract refers to the in silico screen using the random forest model, please make that clear, and also say how many are known to have been screened experimentally.

We edited the abstract to reflect these uncertainties: “We applied htFuncLib to the GFP chromophore-binding pocket, and, using fluorescence readout, recovered >16,000 unique designs encoding as many as eight active-site mutations”.

We added a sentence to the Results saying that the transformation was efficient (estimated at 5×10^7 transformants) and more detail to the Methods section describing the deep sequencing procedure.

1c.

“By eliminating incompatible active-site mutations, htFuncLib generates a greater diversity of functional sequences than evolutionary or mutational scanning approaches for optimizing enzymes, binders, and other functional proteins.” Fig. 3B illustrates that when only looking at variants containing functional site mutations, i.e. removing most variants from the other sets, then htFuncLib seems to cover a larger space. However, it is not clear from the abstract that the other sets are heavily reduced, in particular because variants with more mutations are likely to have at least one outside the functional site. The text in the conclusion is a bit clearer on this point, but it would be good that the abstract reflects this.

Again, we agree. We removed the comparison with previous strategies from the abstract, replacing it with: “By eliminating incompatible active-site mutations, htFuncLib generates a large diversity of functional sequences”.

We also added statements to the figure legend to clarify that mutations outside the pocket were ignored (“Variants with mutations outside the chromophore pocket were included, but these mutations were ignored when calculating distances.”) and added a parenthetical sentence when introducing this analysis in the Results that the previous libraries we refer to did not focus diversity on the chromophore binding pocket (“(albeit, these studies did not focus diversity on the active site)”).

2)

Major parts of the methods are not described in any substantial detail. For example the deep sequencing and random-forest model which are described with six and three sentences respectively. This makes it almost impossible to judge the results and conclusions of the paper, and for others to build on this work. Ideally, one should be able to examine all major parts from reading the paper, for example, only details should be looked up in the code.

Right. We added extensive detail to both sections.

3)

I find it surprising that PROSS-eGFP performs relatively poorly in the computational and experimental stability evaluation (Fig. 5a). If I understand correctly, PROSS-eGFP should be optimized by something very similar to the filtering method so why does so many beneficial substitutions show up in the filtering? The only information I can find regarding the “filtering” is that this leaves a combinatorial space of 10^{18} variants; it would be interesting to see the resulting substitutions presented as in supplementary table S1. Also, Y145F is described in ref. 33 as a part of PROSS-eGFP but here reintroduced as a new mutation? Similarly, T167I is presented as a new substitution here, but seems to already be in the background in ref. 33 (looking at the DNA sequence below the supplementary table S2). Also, in ref. 33 PROSS-eGFP is reported to have 11 or 12 substitutions relative to eGFP but here only one (interpreting undescribed tics in Fig. 5A and the caption of Fig. 3: “PROSS-eGFP (and eGFP, which are nearly identical in the designed positions)”. All in all, this is somewhat unclear from reading the paper. Maybe it is just me having a hard time finding the data, but then that might also be the case for other readers. It would have been useful to report exact sequences of eGFP and PROSS-eGFP (and sfGFP) and substitutions used. Also, please comment on the apparent low stability of PROSS-eGFP and the relationship to the fact that it is optimized.

We agree that this point needs clarification. The PROSS-eGFP design was tested and shown to have higher resistance to thermal denaturation, but surprisingly, in our study, *functional* thermostability is lower:

- We added the following sentence to the first results paragraph: “Because active-site mutations may reduce protein stability, we chose as a starting point a previously designed version of enhanced GFP, PROSS-eGFP, that exhibited elevated resistance to thermal denaturation³³. In this design, active-site positions, except Tyr145Phe and Thr167Ile, were immutable. In applying htFuncLib, we also allowed design in these two positions.” to clarify both the discrepancy regarding PROSS positions and T167 and Y145.
- We added supplementary Tables 3-4 describing the identities examined for each position at the various stages of the htFuncLib algorithm.
- All control sequences are reported in Supplementary Table 12 (previously Supplementary Table 10).
- Added a clarification for the low functional thermostability of PROSS-eGFP: “We noticed that the PROSS-eGFP parental design is less stable than eGFP when functional thermostability is measured (Figure 5A) rather than thermal denaturation as in the PROSS-eGFP design study³³. Apparently, the PROSS-eGFP design is more resistant to heat denaturation, but its fluorescence is more sensitive to heat than eGFP.”

4)

Regarding the number of tested designs, I cannot make the numbers in supplementary table S5 match the attached csv file: The csv holds 72 sequences incl. 3 controls, i.e. 69 tested designs that all seems to be functional, whereas the text reports 68 designs tested in total? There may be a way to add this up but I could not find it. The 19 variants in sfGFP background are listed in the csv but not in table S5. Also, I can only see 15 functional random-forest designs and 10 AmCyan variants selected for spectral shift in the csv but 16 and 14 are reported in table S5? Most importantly, please report the sequences and number of mutations of the failed design (e.g. in the csv and/or in the text) as this is essential information for the community. Also, please describe in the paper the nine “Additional designs” listed in Table S5 as selected from deep sequencing, of which only one is functional. Again, apologies if these were presented somewhere, but I could not find it.

We thank the Reviewer for the detailed analysis. We indeed had some discrepancies in the table. We updated the supplementary tables detailing all the designs (now Supplementary Tables 11 & 12) and the text to reflect the results more accurately. We added all the nonfunctional sequences to the tables, including their source.

5)

In the top plot of Fig 3A, it seems that 100% of single variants are functional – perhaps this is WT, i.e. zero mutations, and the x-axis is shifted? If this is the case, the RF model seems very optimistic in identifying function with almost 100% single mutants functional (currently $x=2$) or at least more than NGS. With this, I find it odd that the bottom plot shows slightly more functional single variants in NGS than in RF (assuming most single mutants are observed in NGS). Please check and clarify.

Right. We corrected this. The NGS data in figure 3B reflect both libraries, whereas RF refers to nohobnds only (it was only trained and applied to the nohobnds sequence space). This is the reason why the NGS data has more single point mutants than the RF data.

6)

Most striking in Fig 3A is the high fraction of the “16,000 potentially active designs” with more than four mutations. This should be validated better if authors wish to report 16,000 functional variants identified in the abstract. First, the authors only report tests of designs up to 8 mutations whereas a substantial fraction of functional NGS variants have more than 8 mutations. Please report the success rate in experimental tests per number of mutations (also related to pt. 4 above on information about the failed designs). Second, please comment on the statistics in supplementary Table S2: The false-positive rate is 1/12, i.e. ~8% are the falsely predicted functional out of the actual non-functional. This is a quite high number since the paper reports ~90% non-functional (Fig. 2C), i.e. with 100,000 non-functional variants, 8,000 are expected to be false positive. This reflects the imbalance in training on a high-prevalence set (most functional) and applying to a low-prevalence set (fewest functional). Third, there is very little indication of the uncertainty in the NGS experiment. It requires a very deep sequencing to cover 100,000 unique variants without paired-end reads, to an extent that warrants calculation of enrichments. To calculate enrichments, the authors need to have a good idea about the

abundance of a variant prior to selection and there is no discussion on how this is addressed. Please comment on this, e.g. size of transformed library, how many cells are expected to have more than one plasmid, FACS coverage (cells sorted per library size), sequencing coverage (average number of reads per unique variant), which region of GFP is sequenced (maximum 600 bases are sequenced), are all functional variants observed in the non-sorted sequencing, are pseudo-counts applied, frequency of unexpected substitutions and how these are handled, etc.

We replaced the claim on variants with >8 mutations with one focusing on variants with up to 8.

We agree that the NGS coverage is problematic due to the length of the diversified region in the amplicon. Transformation efficiency was $>5 \times 10^7$ for both libraries, as calculated from plating experiments giving us at least fivefold coverage (now noted in the Results). In order to cover the libraries to the fullest, we sorted 10 times the total library size, i.e. $\sim 10^8$ and $\sim 10^7$ for nohbonds and hbonds, respectively (in Methods). As the total number of reads was fairly low, the number of reads after filtration was low as well. We thus used the relatively lax criteria of enrichment > 1 . To lower the false-positive rate, we sorted the libraries twice, purifying and re-transforming the plasmids in between. Unexpected substitutions are handled by the LAST algorithm, which considered fastq quality assessments. We also added supplementary figure 6 that shows the number of reads all sequences in the sorted samples had in the deep sequencing data.

As the nohbonds library is larger than the total number of counts available in the deep-sequencing kit we used, we did not expect to cover the unsorted populations entirely. Therefore, we considered any sequence from the sorted samples that was not represented in the non-sorted sample as enriched, effectively applying a low pseudo-count.

These details are now in the methods section.

7)

It would be interesting to see some more details on the construction of the neighbourhoods. E.g. a supplementary table listing the sites of the filtered mutations could also list the neighbourhood of each site and the number of calculated multipoint mutations. Are these mostly double mutants or higher order mutants?

We added supplementary tables 1-2 (one for each library) detailing which positions were in each neighborhood, how many and which mutations were examined for each position and the complexity of the putative libraries.

8)

It would be appropriate to make some quantitative comparison with the previous version of FuncLib (ref. 24), e.g. by the success rates obtained in experimental validations.

FuncLib and htFuncLib are not directly comparable given the very different premises of the two algorithms. Also, FuncLib's success rate (and htFuncLib's as well) are a complicated and unpredictable function of the number of positions, their sensitivity to mutation and their evolutionary conservation. We therefore do not feel comfortable comparing success rates between these methods. Nevertheless, in several cases we noticed that between $\frac{1}{3}$ and $\frac{1}{2}$ of

the FuncLib designs were active. Here, for small libraries (up to 50,000 variants) the success rate is 10% — lower (Fig. 3C), but extremely high compared to standard diversification methods.

9)

It would be interesting to know with a bit more detail on the phylogenetic analysis. The authors write “In this selection step, we keep mutations that are likely to be present in the natural diversity of sequence homologs and that are moreover predicted not to destabilize the protein native state according to atomistic design calculations³⁵”. GFP is sometimes considered not to have very many natural homologs and fpbase.org (ref 35) contains a lot of synthetic variants. Please give the number of sequences in the phylogenetic analysis and, if possible, indicate how many of these are natural, e.g. belongs to a reference genome.

For the phylogenetic analysis, we only used synthetic variants based on avGFP. The analysis started with 136 such sequences, and filtered these to 53. We added these details to the appropriate methods section: “A total of 153 sequences were retrieved from FPBase, all synthetic variants of avGFP”.

10)

The methods section describes “An alternative mutation selection approach that uses Integer Linear Programming” which is only briefly referenced in the text. This should either be removed or the authors should show the results.

We removed this section.

11)

In the paragraph starting with “Our working hypothesis is that epistatic interactions most frequently arise from three molecular sources (Supplementary Figure 1)” the third point is unclear and not illustrated in supplementary Fig. S1: “(3) stability-mediated interactions caused by the nonlinear relationship between the free energy of folding and the fraction of natively folded and functional protein”.

We added a panel to Supplementary Figure 1 with a schematic visualizing stability mediated epistasis. We also clarified the text describing it in the results section: “stability-mediated interactions in which destabilizing mutations do not exhibit phenotypic differences when introduced singly but reduce stability or expression levels when combined”.

Minor points

1)

It would be helpful if Fig. 1 more directly illustrated what “filtering” and “EpiNNet enrichment” means and where in the pipeline it is performed

We clarified the title of panel D “Apply EpiNNet to select a sequence space enriched with mutually compatible mutations”.

2)

Should T65S be in supplementary Fig. S2? It would be useful for the discussion in Fig. 4

There is no straightforward way to model T65S as it is a part of the chromophore (a noncanonical part of the structure). Furthermore, a Ser would be obscured by the larger Thr.

3)

Fig. 4A caption “GFP488/53” should be “GFP488/530”

Corrected. Thanks!

4)

In methods section under FACS sorting: “E. cloni” should probably be “E. coli”, though I quite like the name “cloni”

Cloni is the commercial name of the bacteria we used. The text was clarified to reflect this.

5)

In the introduction the authors write “and functional multipoint mutants are exceptionally rare”, but do not provide a reference to this general statement. Similarly with the statement “Epistasis is a key reason for the low tolerance to multipoint active-site mutations.”

We added the following sentence to the introduction, alongside an appropriate reference to clarify this: “Epistatic interactions between mutations can severely restrict the chances of finding functional multipoint mutants in an active site”.

Reviewer #2 (Remarks to the Author):

In this study, Weinstein and colleagues use a combination of energetic modeling and high-throughput screening to identify GFP variants with multiple mutations, addressing the challenge of potential negative epistasis between mutations reducing the hit-rate. htFuncLib was used to design a set of point mutations and then combinations of mutations that were energetically favorable. Then, a machine learning EpiNet model was trained to discriminate favorable and unfavorable combinations of mutations. Hits from this approach included those with > 8 mutations, which exceeded the tolerated mutational perturbation load from previous design approaches. This work and a companion submission on enzyme engineering show that issues with epistasis can begin to be addressed by combining judicious energetic modeling combined with training of machine learning models. An important and relevant study to the protein engineering field. The work is technically sound and clearly presented.

We thank the Reviewer for these positive comments.

Two comments that should be addressed:

(1) There is no functional goal in these libraries - i.e. quantum yield, photostability, color. How would these methods be adapted if a particular functional feature, not just structure and stability were to be optimized. Excitation and emission spectral properties were not described (peak

wavelengths). Can models be trained to identify what features contribute to photophysical properties?

This is very true. We explain that htFuncLib's goal (as implemented here) is to generate tolerated sequence diversity. If a particular functional goal is known, and it is understood what molecular details may lead to this goal, then htFuncLib can be adapted to it. For example, if it is known that shorter excitation wavelengths require an aromatic amino acid at a specific position, one may force that mutation and use htFuncLib to recommend mutations that are compatible with it. We added a segment discussing this to the conclusions: "If a specific functional goal is desired and the molecular underpinnings of that goal are known, they can be imposed during the design process to focus the library on variants that exhibit the essential molecular features."

(2) This training of EpiNNet should be discussed in the context of the choice of host protein - a stable version of GFP - and specifically the work earlier this year from Kondrashov (<https://doi.org/10.7554/eLife.75842>) showing that mutational landscapes that are flatter are not as useful for training models. Does the energetic modeling in htfuncLib work for more 'fragile' proteins where epistatic interactions can have a more pronounced effect on folding/function?

Right. We do not yet know the answer to this, though it is likely that protein robustness is important for success. We added a sentence to the Conclusion section discussing this: "The high stability and brightness of the eGFP starting point are likely to be key to obtaining a large number of functional variants. Further research is needed to determine whether the combination of PROSS stability design and htFuncLib can access such large spaces of functional variants in less robust starting points".

Reviewer #3 (Remarks to the Author):

In this manuscript, the authors introduce the htFuncLib, a protein-engineering pipeline to design and test variant libraries focused on protein functional sites. The motivation behind developing such a method is to increase the sampling efficiency and diversity around a protein functional site, which is usually highly conserved and sensitive to mutations. To achieve this goal, the authors have to overcome the epistatic effect by introducing multiple mutations simultaneously, which, in the past, has only been partially achieved by directed evolution through iterative mutation-selection cycles. The htFuncLib method starts with low-energy PSSM-approved mutations in the functional site (by phylogenetic analysis and Rosetta energy calculation), then ranks and selects those mutations by their mutual compatibility (by a trained neuron network EpiNNet). After this in silico screening, DNA fragments encoding these compatible point mutations are assembled in an all-against-all combinatorial library by Golden Gate method, tested by high-throughput FACS, and read out by deep sequencing. The authors apply this engineering pipeline to GFP's fluorescence functional site. The results are impressive: 1.) they explore a much bigger sequence space that is inaccessible in multiple previous attempts, 2.) the functional multipoint mutants after library selection show desired functional diversity in terms of protein stability, fluorescence spectra, fluorescence lifetime, pH sensitivity, and fluorescence

photo-stability, and 3.) the molecular mechanisms of epistasis underlying the successfully selected GFP variants are interesting for structural analysis. Overall, the manuscript presents a pragmatic way to diversify certain protein functions and I anticipate it will attract attention among protein engineers working towards protein tools (eg. imaging tools such as fluorescent proteins) and enzymes, thus I recommend this manuscript for publication after a minor revision.

We thank the Reviewer for the positive assessment.

Below are my comments for the authors:

1. A direct comparison between FuncLib and htFuncLib would be necessary here. If adding a perceptron-based neural network (or ILP) machine learning module largely improves the end results, it would be worthwhile to ask what Rosetta method lacks and what role Rosetta design calculation plays in this new method.

FuncLib and htFuncLib have different goals. The first searches for a small set of optimized mutants in which each design is independent of any other and the second for a large set of mutants that can be assembled by combinatorial mutagenesis. As such, htFuncLib does not improve FuncLib but uses it as a platform to extend to much larger sequence spaces. We do not think that the results we show directly suggest a lack in Rosetta. If anything, Rosetta is the engine for computing the compatibility among mutations.

2. The general applicability for other users and other proteins is not very clear. There are several manual steps in the Method description (Line 455, 462, 470). While it is understandable to introduce manual intervention on every steps during method development and the initial application, I would like to see how the authors plan to automate the pipeline for future applications.

We agree. Our lab is committed to making design tools that are streamlined and accessible for researchers with no expertise in modeling. We are currently working on an “upgrade” to the FuncLib web server that will enable automated or semi-automated execution of htFuncLib. We hope to have this in ready form within the next six months.

3. The final paragraph in the Introduction is slightly an overstatement (Line 55, “arbitrarily large libraries” and Line 61, “millions (and potentially billions) of designs”). From the Method description, it is obvious that the size of the final library is a limiting factor for designing the combinatorial library (Line 462-464, Line 494-497). I would suggest the authors to revise this paragraph to avoid misleading.

We think that the method is scalable well beyond millions but certainly do not yet have data to support that. We therefore eliminated the claim on billions of designs.

4. In Line 81-86 and Supplementary Figure 1, the authors listed three hypothesized sources for epistatic interactions. It is hinted in the text (Line 99, “penalize backbone deformation” and Line 105 “most likely to give rise to...”) that the htFuncLib is focused on establishing type 1 epistatic interactions only (this is my speculation). It would make the manuscript easy to understand if the

authors can offer a direct correspondence between the three types of epistatic interactions and the htFuncLib library design's target interactions.

We edited the text to reflect how the algorithm addresses the three types of epistasis we mention:

The end of the third paragraph in the results section describes how indirect, backbone mediated, epistasis is addressed: "In addition, these calculations apply harmonic coordinate constraints to backbone atoms during whole-structure minimization, thereby penalizing backbone deformations that may lead to indirect epistatic interactions".

The following paragraph states that: "Since the space of potential multipoint mutations in a large active site is computationally intractable, we focus calculations on combinations of mutations within neighborhoods of proximal positions (Figure 1B & C, Supplementary Tables 1 and 2) which are the most likely to give rise to direct epistatic interactions (Supplementary Figure 1A)", accounting for how the algorithm addresses direct epistasis.

The same paragraph later states: "The resulting library is enriched in mutually compatible mutations, such that both direct and stability-mediated epistasis (Supplementary Figure 1A and C) are addressed".

5. Following comment 4, I am also confused about how the method deals with backbone movement upon introducing multiple mutations, eg. how does the calculation "penalizes backbone deformation(Line 105)"?

We edited the text to better explain this point: "In addition, these calculations penalize backbone deformations to minimize indirect epistatic interactions by applying harmonic coordinate constraints on backbone atoms (Supplementary Figure 1B).".

6. In Fig.2D, the overlay of the top-ranked mutations could be better illustrated in a different color.

We changed the colors of the various mutations.

7. In Fig.3A, "NGS" could be better named as "htFuncLib-NGS". I misunderstood it as all the next-generation sequencing data combined (or, does it really mean all the data combined? See, I'm confused.).

We changed the figure accordingly.

8. In Fig.3A, it does not make clear sense to me that the point mutants ("1" in the bottom plot) have a functional ratio of 100% ("1.0" in the top plot) for all the libraries and "RF" prediction. Is it a normalization point? If it is not a normalization point, does it indicate that the fluorescence threshold for defining "functional" is arbitrarily low in this analysis? In addition, since the other reference libraries (avGFP, cgreGFP, ...) are sorted differently, I wonder how to justify this comparison of "functional" variants.

Corrected.

9. Fig.4 and Supplementary Fig. 8 are the same low-dimensional representation of protein fitness landscape labeled with "clean" Random Forest(RF) predicted functional mutations and "noisy" experimental data, respectively. The authors choose to focus their analysis and

discussion on the RF-predicted results (Fig.4) in the main text. While this is totally reasonable with proper justification (as the authors have provided in line 355-356 for "false negative" and line 885-886 for "false positive"), it should be noted if the representative mutations have strong or weak signals in the experimental data. If they are completely missed in the FACS sorting and NGS sequencing, further experimental validation is needed to support the authors' claim. For example, the discussion on the "two long parallel tails" in line 323-344 is not very convincing to me since the same signals are not apparent in Supplementary Fig. 8A. To keep this part as a novel finding, I would suggest the authors test the representative mutations experimentally.

We have added information in the main text and Figure S8 about the direct support in the experimental data that each of the different clusters of sequences have in the experimental data

- For the three main connected clusters differing at positions 65 and 69: "All three groups are strongly supported by many different sequences directly assessed in the sorting experiment (Fig S8)"
- We clarified that the experimental data strongly supports the AmCyan "tail": "The AmCyan^{405/525} tail is well-supported in the experimental data and is not an artifactual prediction of our model, as we observe a cluster of highly mutationally connected designs that were also among the most strongly enriched in AmCyan^{405/525} sorted cells (Figure 4D, Figure S8)"

To make these results more transparent, we provide a new Supplementary Table 10 with the sequences belonging to the cluster and their corresponding enrichment values as shown here:

L42	V68	Q69	S72	T108	V112	Y145	T167	H181	L220	V224	Functional class	Enrichment (log2)
V	A	A	T	E	V	Y	T	H	V	I	AmCyan	7.2
V	A	A	T	E	V	M	T	H	V	I	AmCyan	7.6
V	A	A	T	E	V	F	T	H	V	I	AmCyan	5.2
V	A	A	T	E	V	Y	V	H	V	I	AmCyan	7.8
V	A	A	T	E	V	Y	T	H	L	I	AmCyan	6.2
V	A	A	T	E	V	I	T	H	V	I	AmCyan	5.2
V	A	A	T	E	V	Y	T	L	V	I	AmCyan	5.2
V	A	A	T	E	V	M	V	L	V	I	AmCyan	5.2
V	A	A	T	E	I	Y	T	H	V	I	GFP	1.3

10. Reference data of transferring mutations to sfGFP are missing (Line 361-362).

Added a reference to the appropriate supplementary table.

11. For the 68 unique designs (Line 349) chosen for protein purification and biochemistry characterization, how many functional-site mutations do they carry?

We added a comment to clarify this: “exhibiting at least two mutations from PROSS-eGFP and typically at least two mutations from one another”.

12. An open and honest discussion on the limitations of the htFuncLib method would make this manuscript stronger. From several places in the main text, the htFuncLib seems to require a very stable starting point and it cannot explicitly improve a specific aspect of the protein function. I think that general readers will appreciate an open discussion in this regard.

We edited the paper in several places to reflect the potential limitations of the method, including:

- A paragraph in the conclusion section which states: “The high stability and brightness of the eGFP starting point are likely to be key to obtaining so many functional variants. Further research is needed to determine whether the combination of PROSS stability design and htFuncLib can access such large spaces of functional variants in less robust starting points. “
- A statement regarding the apparent stability of our starting point in the results section: “We noticed that the PROSS-eGFP parental design is less stable than eGFP when functional thermostability is measured (Figure 5A) rather than thermal denaturation as in the PROSS-eGFP design study. Apparently, the PROSS-eGFP design is more resistant to heat denaturation, but its fluorescence is more sensitive to heat than eGFP”.
- A paragraph in the results section that states: “Because active-site mutations may reduce protein stability, we chose as a starting point a previously designed version of enhanced GFP, PROSS-eGFP, that exhibited elevated resistance to thermal denaturation”.

Reviewer #4 (Remarks to the Author):

The manuscript describes development and application of the htFuncLib – a computational protein design workflow combining atomistic and machine-learning based approaches. The goal of the htFuncLib is to increase efficiency of laboratory screening efforts by eliminating poorly scoring combinations of mutations from combinatorial libraries, allowing exploration of highly epistatic fitness landscapes. Even for a limited set of manually curated designable positions, an astronomically large number of combinations makes exhaustive sampling of the full sequence space computationally intractable. To optimize amino acid composition for each position the authors first used phylogenetic information and in silico site saturation mutagenesis (SSM) to identify residue types most likely individually tolerable in the parent sequence context. However, as their computations show, a dominant fraction of the designs constructed by random combination of these individually beneficial mutants has substantially worse computational score relative to parental sequence.

The authors propose a simple and elegant computational procedure that apparently helps to alleviate this problem. The entire set of designable positions is split into spatial neighborhoods and the range of allowed amino acids for each position is flexibly adjusted so as to make the total number of sequence combinations for the neighborhood computationally tractable (under 10^6). Rosetta energy score is computed for each combination (or 10% of all combinations for large neighborhoods) in each neighborhood and combinations are classified as “good” or “bad” relative to the score of the parental sequence. Authors train a neural network (EpiNNet) to classify designs, and use the trained network to rank individual mutations in a way reflecting probability of the mutation to be in a “good” scoring combination. The main discovery of the study is that designs constructed from higher ranking mutations (“EpiNNet enriched”) have a much higher chance to have better energy scores than designs constructed from a set of mutations filtered using phylogenetic information and single SSM computations.

We thank the Reviewer for the positive assessment.

Authors proceed to apply their workflow to construct a library of PROSS-eGFP – a previously optimized variant of avGFP. Sorting library using FACS indicated presence of the variants retaining fluorescence even with up to 8 mutations. While the functional status of the overwhelming majority of >16,000 variants was assigned based on their enrichment in NGS data, some variants were purified on a scale sufficient for more detailed biophysical analysis.

While the rationale of the method is well laid out and convincing, the findings seem a bit underwhelming.

First, it appears that library construction using high ranking mutations lead to a mostly very conservative set of allowed mutations. Considering this, the result of finding functionally active variants with up to 8 mutations in the active site becomes almost trivial. Authors remark on the inclusion of radical substitutions into the library, but it is not clear how often such mutations appear in the active variants and more importantly how often presence of the radical substitutions affect functionality in a practically significant way. Additionally, previous studies referenced in the manuscript (Somermeyer, 2022) indicated various levels of robustness for different variants of the GFP. avGFP was reported to have intermediate robustness (tolerating at most 4 mutations), but given extensive optimization of PROSS-eGFP it may be not surprising its robustness increased to the level allowing it to tolerate more mutations. It might be helpful to see how the method performs for less optimized proteins.

Actually, the libraries introduce many radical mutations, as can be seen in Figure 2D, supplementary Figures 2 and 3, and supplementary Table 5. It is also briefly mentioned in the text: “Both libraries are complex: some positions allow only subtle mutations, and others, including e.g., Gln69 and Tyr145, exhibit high diversity and radical mutations (Supplementary Figures 2 and 3, Supplementary Table 5)”.

Additionally, we added the following text and appropriate table: “Strikingly, many of the active designs have radical mutations, including Thr203His (13%), Gln69Met (9%), Ser205Asp (9%), Gln94Leu (8%), and Tyr145Met (8%) (Supplementary Table 7).”

Second, it would be helpful to see a control experiment where a library is created by combining lowest ranking mutations, or selected positions completely randomized to be able evaluate significance of the sequence space optimization provided by EpiNNet or any other method.

The analysis depicted in Figure 2C shows that once lower ranking mutations are added to the synthetic library, the success rate falls. By extrapolating this trend, and the fact that the non enriched sequence spaces performed extremely poorly in the in silico energy analysis (Figure 2A) we conclude that the bottom ranking mutations are likely to encode a library that is depleted in functional variants.

Third, authors claim functional variants having a wide range of biophysical properties (spectral changes, thermostability changes, quantum yield, photostability, life time etc.). Given the multi-modality of fluorescent function and its sensitivity to the immediate environment it is not surprising to discover variants have functional diversity. It is exciting to see few variants with greatly increased thermostability or fluorescence lifetime, but on the other hand almost all variants are less photostable making their use limited to specific applications. And yet it is hard to imagine similar results cannot be obtained with other types of diversification as exemplified by multiple examples of directed evolution experiments with fluorescent proteins. Interestingly, it appears none of the characterized variants have substantial changes in emission spectra, which is quite often the most desired feature to be modified.

PROSS-eGFP is a very high bar given its high optimization, making the significant improvements observed in some of our designs quite striking, in our opinion. The key point of the biochemical analysis is that choosing designs from the FACS-selected population (almost arbitrarily) leads to a very high diversity in multiple functional properties; the designs do not merely expose “mutational tolerance” as most previous studies did, but potentially useful functional diversity. As different fluorescent characteristics are required for different experimental procedures, we suggest follow up studies use our sorted libraries (deposited in AddGene) to screen for specific characteristics, such as high photostability.

We also note that not all applications require the spectral parameters to go in a certain direction. For instance, low photostability is useful in FRAP experiments whereas high photostability is crucial for long term experiments (this is mentioned on pg. 13). Stability in different pH also does not exhibit an “optimum” in the usual sense of the word. The bottom line is that htFuncLib currently provides the only way to generate functional diversity that may be relevant to multiple, even conflicting, optimization goals. We added a statement to the conclusions section discussing this: “Our implementation of htFuncLib did not target a specific functional outcome, except for protein stability. This implementation is especially suitable if multiple variants for different and potentially incompatible goals are desired. For example, FRAP experiments require fluorescent proteins that bleach quickly, whereas long-term imaging experiments require slow bleaching, and we recovered designs that exhibited both properties from a single library.” We believe that the comment the Reviewer makes on directed evolution misses the point: it is certainly true that directed evolution leads to improvements, but typically a laborious, multi step mutation/selection process is required *for each functional goal*. Here, we found thousands of potentially useful sequences from a single experiment.

Regarding emission spectra, those are mostly modified by the residue at position 66 (within the chromophore), which we did not modify in the current study.

Fourth, it is not clear how significant is the role of EpiNNet in ranking and selection of the mutations to be allowed in the library. Alternative methods to perform this task were described and applied to optimize enzymes (Fox, 2005 DOI: 10.1016/j.jtbi.2004.11.031; Fox 2007 <https://doi.org/10.1038/nbt1286>)

The htFuncLib approach requires no experimental data (other than a structure and a multiple sequence alignment of homologs), whereas the ProSAR approach requires multiple iterations of experiment and analysis. htFuncLib is essentially a single shot method for introducing multiple active site mutations.

On a subject of presentation.

Lines 204-205 Describe how many libraries were constructed, and were there more than just nohbonds and hbonds libraries.

No, we only tested these two.

Lines 213-216 Reference 20 used epPCR to construct libraries with no ability to target mutations specifically to chromophore-binding pocket, so there were probably no mutants with ≥ 5 mutations in the active site

Right. We added a statement regarding this to the appropriate paragraph: "albeit, these studies did not focus diversity on the active site".

Lines 267-269 How conservation score is defined and deltaPSSM is computed?

We edited the appropriate methods section to clarify this: "In addition, for every variable position, the difference in the surface accessible solvent area (SASA), PSSM score, and amino acid category were also assigned (comparing the mutated amino acid and the PROSS-eGFP identity)".

Lines 494-496 Not clear what role EpiNNet plays in selecting mutations for the library. It appears the EpiNNet is used to rank the mutations, but not select them. Therefore, comparison between ILP solution and EpiNNet mutation-selection process is confusing and misleading

We chose to remove the ILP discussion from the main text and methods to reduce confusion. Indeed, EpiNNet ranks the mutations and the user selects the number of top-ranked mutations according to the library size s/he wants to construct.

Line 459 should it be scores greater than -2 ?

Yes. corrected in the text. Thanks!

Lines 462-463 Methods section describing "Refinement and mutation scan" seems to contain text relevant for "Partial modeling and scoring".

Correct, we moved this sentence to the correct position.

Lines 114-115 seems to erroneously reference an ILP solution to picking ranked mutations as a method to train a model to perform classification task.

We removed all mentions of the ILP process.

Line 584 ... and purified using <what?>

We corrected the text.

Line 618 should be pET28 instead of pBAD?

Correct, we edited the text accordingly: " Genes were inserted in the pET28 vector using BsaI restriction sites cloned previously using QuickChange".

Line 619 ... restriction sites cloned introduced previously

Corrected to "Genes were inserted in the pET28 vector using BsaI restriction sites previously cloned using QuickChange".

Line 625 BL21(DE3) cells not BL21 cells? Expression from pBAD vector can be done in BL21 cells, pET vectors with T7 promoters require BL21(DE3) cells

We corrected the text to: "pET28 plasmids containing the relevant insert were transformed into BL21 (DE3) cells and grown overnight."

Figure 3A top panel suggests all # mutations=1 are active in all libraries, which is probably incorrect. Also the color scheme is illegible, please make markers of different shapes to help read the graph.

We corrected both the register for the entire figure and add markers.

Figure 1C What is the side-bar color gradient illustrating?

We edited the caption to explain this: "Color bars signify increasing Rosetta energies."

Page 3 line 105 and page 15 line 469 A unit of distance, presumably Å, is missing. There is also inconsistency in font on page 15.

The issue was with the system's PDF, and was corrected.

Page 3 The definition given in parentheses of proximal positions is incomplete and misleading. Perhaps use definition given in methods which is much more effective.

We removed the definition from the results section, in favor of the detailed explanation in the appropriate methods section.

Page 5 The hbond and nohbond libraries differ in their definitions only in their inclusion of positions involved in hydrogen bonds to the chromophore, but the smaller number of positions included in the hbond library indicates some additional criteria differentiating selection of positions in these two libraries

By starting with a different set of positions, we ended testing and choosing a different set of positions to actually mutate.

Page 52 Supplementary table 6 appears to be missing.

Supplementary Table 6, now 10 was an additional file.

Page 12 line 369 Functional thermostability values are missing units, presumably °C.

Corrected.

Reviewer #1 (Remarks to the Author):

The revised paper by Weinstein et al improves the manuscript and removes some of the claims that were previously not well supported by the data in the original version. For example, a sentence in the abstract that originally read:

"We screened 11 million htFuncLib designs that targeted the GFP chromophore-binding pocket and isolated >16,000 unique fluorescent designs..."

now reads

"We applied htFuncLib to the GFP chromophore-binding pocket, and, using fluorescence readout, recovered >16,000 unique designs"

It's still a bit unclear what the 16000 unique designs are and how many of them are fluorescent.

For example, in the first review, I had asked (pt. 6) about the ~8% false-positive rate, which might correspond to 8000 designs. This point still remains to be answered. On the other hand, a careful reader might find this number in Table S2.

Reviewer #2 (Remarks to the Author):

The author responses to my questions were adequate and do not require further revision. Out of larger scientific interest beyond the scope of this study, I mention the following points:

With regard to the first question about selection for specific functional features (in the case of a fluorescent protein - color, quantum yield, etc), the authors indicate that knowledge about chemical/physical constraints could be incorporated into the design scheme. This is fine, although it would be more exciting to learn these features through selection of a large, negative-epistasis free, library.

Regarding a fragile vs. robust starting point - this approach may help us better understand the possible anticorrelation between a large space of sequence designability in a robust background versus a smaller space in a fragile background that could yield greater functional innovation.

Overall a very exciting study and an important contribution to the field.

Reviewer #3 (Remarks to the Author):

In the revised version of their manuscript, the authors have faithfully addressed the majority of my questions and concerns. I recommend the manuscript for publication.

Reviewer #4 (Remarks to the Author):

I would like to thank the authors for their clarifications and changes made to the manuscript. I do not have more questions and would recommend publishing the work.

On behalf of all authors, we would like to sincerely thank the reviewers for their thorough and thoughtful reviews. We feel the manuscript is much better off thanks to their help.

Reviewer 1:

Reviewer #1 (Remarks to the Author):

The revised paper by Weinstein et al improves the manuscript and removes some of the claims that were previously not well supported by the data in the original version. For example, a sentence in the abstract that originally read:

“We screened 11 million htFuncLib designs that targeted the GFP chromophore-binding pocket and isolated >16,000 unique fluorescent designs...”

now reads

“We applied htFuncLib to the GFP chromophore-binding pocket, and, using fluorescence readout, recovered >16,000 unique designs”

It's still a bit unclear what the 16000 unique designs are and how many of them are fluorescent.

For example, in the first review, I had asked (pt. 6) about the ~8% false-positive rate, which might correspond to 8000 designs. This point still remains to be answered. On the other hand, a careful reader might find this number in Table S2.

We revised the legend of table S6 (now supplementary table 1) to better explain this.

Reviewer #2 (Remarks to the Author):

The author responses to my questions were adequate and do not require further revision. Out of larger scientific interest beyond the scope of this study, I mention the following points:

With regard to the first question about selection for specific functional features (in the case of a fluorescent protein - color, quantum yield, etc), the authors indicate that knowledge about chemical/physical constraints could be incorporated into the design scheme. This is fine, although it would be more exciting to learn these features through selection of a large, negative-epistasis free, library.

Regarding a fragile vs. robust starting point - this approach may help us better understand the possible anticorrelation between a large space of sequence designability in a robust background versus a smaller space in a fragile background that could yield greater functional innovation.

Overall a very exciting study and an important contribution to the field.

Reviewer #3 (Remarks to the Author):

In the revised version of their manuscript, the authors have faithfully addressed the majority of my questions and concerns. I recommend the manuscript for publication.

Reviewer #4 (Remarks to the Author):

I would like to thank the authors for their clarifications and changes made to the manuscript. I do not have more questions and would recommend publishing the work.